# The Differential Expressions and Associations of Intracellular and Extracellular GRP78/Bip with Disease Activity and Progression in Rheumatoid Arthritis

**DOI:** 10.3390/bioengineering12010058

**Published:** 2025-01-13

**Authors:** Guoyin Liu, Jianping Wu, Yongqiang Wang, Yuansheng Xu, Chun Xu, Guilin Fang, Xin Li, Jianmin Chen

**Affiliations:** 1Department of Orthopedics, The Affiliated Jinling Hospital of Nanjing Medical University, Nanjing 211166, China; liuguoyin0425@163.com (G.L.); nanjingxys2014@163.com (Y.X.); 2Department of Orthopedics, The First Affiliated Hospital of Nanjing Medical University, Nanjing 210029, China; 3Department of Obstetrics, The Affiliated Jinling Hospital of Nanjing Medical University, Nanjing 211166, China; 15951836482@163.com; 4Department of Rehabilitation, The Affiliated Jinling Hospital of Nanjing Medical University, Nanjing 211166, China; jiva0122@163.com; 5Department of Pathology, Nanjing Drum Tower Hospital, Affiliated Hospital of Medical School, Nanjing University, Nanjing 210008, China; xuchun5286@163.com; 6Department of Rheumatology, The Affiliated Jinling Hospital of Nanjing Medical University, Nanjing 211166, China; lgycxl0425@163.com; 7Department of Orthopedics, Central Military Commission Joint Logistics Support Force 904th Hospital, Wuxi 214044, China

**Keywords:** intracellular GRP78/BiP, extracellular GRP78/BiP, rheumatoid arthritis (RA), disease activity, disease progression

## Abstract

GRP78/BiP, a stress-induced protein and autoantigen in rheumatoid arthritis (RA), exhibits different expressions in various biological fluids and tissues, including blood, synovial fluid (SF), and synovium, all of which are pertinent to the disease activity and progression of RA; however, there is a scarcity of data linking both intracellular and extracellular GRP78/Bip to disease activity and progression of RA. This study was undertaken to investigate the differential expression of GRP78/Bip in blood, SF, and synovium, and to determine their association with disease activity and progression of RA. Patients with RA, osteoarthritis (OA), and traumatic meniscal injury (TMI) without radiographic OA were consecutively recruited for the study. Among patients with RA, six different subgroups were established based on their disease activity and progression. Disease activity was measured using the DAS28 (Disease activity scores in 28 joints) criterion, while disease progression was evaluated using the Steinbrocker classification grade. The levels of GRP78/Bip, TNF-α, and IL-10 were significantly elevated in the serum, SF, and synovium of patients with RA when compared to both the control (CON, TMI Patients) and the inflammation control (iCON, OA Patients) groups (*p* < 0.05). In terms of disease activity status, as opposed to remission status in RA, the levels of GRP78/Bip, TNF-α, and IL-10 were all elevated in serum and synovium (*p* < 0.05). However, GRP78/Bip and IL-10 levels were found to be reduced in SF, while TNF-α levels remained elevated. With respect to disease progression in RA, GRP78/Bip levels exhibited a positive correlation with both the stage of RA and the levels of TNF-α and IL-10 in serum and synovium. Nonetheless, a negative correlation was observed between GRP78/Bip levels and the stage of RA in SF, while positive correlations with the levels of TNF-α and IL-10 persisted. The differential expression of GRP78/Bip in blood, SF, and synovium indicated that the potential role and function of GRP78/Bip might vary depending on its specific location within these biological fluids and tissues. The presence of intracellular and extracellular GRP78/Bip was associated with disease activity and progression of RA, suggesting the involvement of GRP78/Bip in the pathogenesis and development of this debilitating autoimmune disorder, as well as its potential as a biomarker for monitoring disease activity and progression of RA.

## 1. Introduction

Rheumatoid arthritis (RA) is a systemic autoimmune disease characterized by chronic inflammation, abnormal synovial proliferation, and the formation of pannus [1]. These manifestations are mainly apparent in synovial joints, ultimately resulting in the destruction of diarthrodial joints [2]. The state of RA progressively worsens, following a pattern of alternating between periods of disease quiescence and progression, as well as disease activity and remission [3]. One key aspect of the development of RA involves the proliferation of stressed cells, while evading immune destruction and apoptosis is crucial [4]. As inflammatory infiltration increases, irreversible destruction of the articular cartilage and bone tissue occurs. Failure to address this promptly in the joint can lead to chronicity with resultant joint damage and deformity [5]. The long-term consequences of the disease are characterized by significant morbidity, loss of functional capacity, permanent disability, and increased mortality, presenting a significant challenge for contemporary society [6].

The treatment of RA has undergone two major paradigm shifts that have improved the quality of life of patients to a previously unthought degree. The first shift occurred with the development of disease-modifying drugs, which not only alleviated symptoms but also slowed down or even prevented joint damage [7]. The second came with the introduction of targeted therapy through the use of biologic agents [8,9]. These biological treatments have transformed the therapeutic landscape for RA. However, despite these notable advancements in available therapies, it remains concerning that a substantial proportion, specifically between 20% and 40% of patients, have been classified as non-responders following the initiation of biologic therapy [10,11]. The high expenses and potential exposure to severe adverse reactions highlighted the critical need of identifying biomarkers capable of differentiating between pre-treatment responders and non-responder patients. Additionally, it is essential to investigate the differential expressions of these biomarkers and their associations with disease activity and progression in individuals suffering from RA.

Current therapies for RA inhibit components involved in the inflammatory disease process without directly facilitating the resolution of inflammation [12,13]. Appropriate regulation and subsequent resolution of acute inflammatory events are essential for preventing autoinflammatory conditions [14]. Indeed, the chronic inflammation that characterizes RA is at least partially a consequence of the inadequate functioning of endogenous immunoregulation [14]. This highlights the importance of not only addressing inflammation but also fostering the body’s ability to effectively resolve it, which is essential for managing RA more effectively and preventing its long-term consequences [15]. The therapeutic goal of rheumatologists is the induction of drug-free remission [14]. Ultimately, of course, prolonged or life-long drug-free remission would be indistinguishable from a cure [16]. Consequently, the next generation of RA therapeutics will complement and augment endogenous immunoregulatory and pro-resolution immunological networks, thus promoting a definitive and long-lasting resolution of inflammation rather than merely temporary immunological control [16,17,18]. Of particular interest regarding this therapeutic approach is the 78 kDa glucose-regulated protein (GRP78), also known as binding immunoglobulin protein (BiP) [19]. In the Phase I/IIA clinical trial conducted on RA, GRP78/Bip therapy demonstrated a safety profile that was both favorable and well tolerated by the participants [20]. Furthermore, the biomarker analysis performed during the trial revealed considerable anti-inflammatory activity, which was accompanied by tangible clinical benefits for the participants [20]. This evidence supports the notion that GRP78/Bip therapy may represent a promising option for the management of RA, thereby warranting further investigation and development [20].

GRP78/BiP, a significant stress protein belonging to the HSP70 family, is encoded by the HSPA5 gene [19]. GRP78/BiP serves as a vital molecular chaperone within the endoplasmic reticulum (ER), playing an essential role in the proper folding and processing of newly synthesized membrane-bound or secretory proteins [21]. In addition to its chaperoning activities, GRP78/BiP is a crucial regulator of the unfolded protein response (UPR) and ER stress [21]. The activation of this stress response is critical for re-establishing equilibrium and maintaining cellular homeostasis as it improves the cell’s ability to manage and eliminate these dysfunctional proteins, thereby reducing the potential damage resulting from their accumulation [22]. Since the identification of GRP78/BiP as the protein that binds to the heavy chains of immunoglobulins, it has been implicated in exerting functions besides chaperoning premature proteins that are of immunologic importance, such as non-classical antigen presentation and regulation of cytotoxic T cell responses [23]. Thus, GRP78/BiP is not only essential for the proper folding and maturation of proteins but also has pivotal implications in the modulation of immune responses [23]. Recent research has classified GRP78/BiP as a member of the resolution-associated molecular patterns (RAMPs) family of molecules [14,24]. This classification underscores its crucial role in various stress responses where it contributes to the provision of immunoregulatory and pro-resolution signals within immune networks [14,24]. Immune mechanisms in RA are regulated by a diverse group of endogenous proteins [25]. These proteins can be broadly categorized into two main groups: pro-inflammatory damage-associated molecular patterns (DAMPs) and anti-inflammatory RAMPs [14]. Nevertheless, unlike DAMPs, the predominant cellular effect of RAMPs is to suppress the secretion of cytotoxins and pro-inflammatory agents by immune cells [14]. This positions most RAMPs as pivotal in modulating the immune response, facilitating the resolution of inflammation and contributing significantly to homeostasis maintenance within the body [24]. Consequently, RAMPs such as GRP78/BiP may represent novel immunotherapeutic approaches capable of inducing drug-free remission in patients with autoinflammatory conditions [14,26].

Notably, GRP78/BiP appears to be a marker of the stress status of the cell, since under stress conditions, it migrates out of the ER into the nucleus and to the cell surface, in addition to being released outside the cell [27]. It is increasingly recognized as a multifaceted protein capable of adapting to different forms of cellular stress through modifications in its localization [27]. This evolution in our understanding represents a transformative change in how we perceive the roles of GRP78/BiP, indicating its integral involvement in the cellular response to stress and the maintenance of cellular homeostasis, thereby influencing a wide array of physiological and pathological processes [28,29,30,31]. Multiple studies have provided evidence of significant differences in the functions of GRP78/BiP, which vary considerably depending on whether the protein is located in extracellular or intracellular environments [28,30,31,32,33,34]. This adaptability and versatility allow GRP78/BiP to be classified as a genuine moonlighting protein, emphasizing its ability to perform a variety of distinct and specialized roles within biological systems [27,29,30,35]. The intracellular form of GRP78/BiP serves as a critical chaperone in protein folding within the ER and exhibits anti-apoptotic and pro-survival properties through both direct or indirect interactions, thereby aiding in cellular protection under stressful conditions. As an extracellular protein, GRP78/BiP demonstrates potent immunomodulatory capabilities and mediates various anti-inflammatory and pro-resolution actions that facilitate the timely resolution of inflammation.

During the pathogenesis process of RA, GRP78/BiP translocates to the cell surface and is secreted into the extracellular environment, functioning as an autoantigen to induce autoantibodies and enhance immune responses. Consequently, GRP78/BiP and anti-GRP78/BiP antibodies have been detected in connective tissue (inflamed synovium) [23,36,37,38,39,40], as well as in several bodily liquids, including salvia [41], blood [14,36,39,40,42,43,44,45,46,47,48,49,50,51], and synovial fluid (SF) [42,44,52,53,54]. Furthermore, the upregulation of GRP78/BiP appears to be distinctive of RA and is independent of drug therapy [20]. These findings indicate that GRP78/BiP could be explored as a potential biomarker to improve diagnostic algorithms and monitor the prognostic effect of RA [20]. However, current research on GRP78/Bip remains fragmented and lacks a comprehensive perspective, as existing studies on RA primarily focus on the role and targeted intervention of GRP78/Bip in blood, SF, or synovium, without adequately considering both disease activity and progression throughout the course of RA. This absence of comprehensive information emphasizes a critical gap in our current understanding of how this protein interacts and exerts its influence within the framework of the disease. Addressing these questions is essential for a more complete perspective on the roles that GRP78/Bip may play in the pathophysiology of RA and could potentially pave the way for a more personalized therapy targeting GRP78/Bip in RA. Thus, the aim of this study was to investigate the differential expression of GRP78/Bip in blood, SF, and synovium, as well as to explore the relationships between the intracellular and extracellular levels of GRP78/Bip and the disease activity and progression of RA. A better understanding of the functions of intracellular and extracellular forms of GRP78/Bip, along with the mechanisms regulating GRP78/Bip-induced immune responses, may lead to innovative strategies for the management of RA and possibly other inflammatory diseases.

## 2. Material and Methods

### 2.1. Ethics Statement

The experiment received approval from the ethics committee of The Affiliated Jinling Hospital of Nanjing Medical University (Approval Number: 81YY-KYLL-13-09). All participants were fully informed about the process and possible adverse effects of the investigation and instructed to report any adverse reaction to researchers during the experiment. All testing was voluntary, confidential, and undertaken with the patient’s written informed consent, and the rights of the participants were protected.

### 2.2. Study Design and Setting

Participants were consecutively recruited over a period from June 2020 to June 2023 through a convenient sampling method. Patients with knee RA were diagnosed based on the 2010 American College of Rheumatology (ACR)/European Alliance of Associations for Rheumatology (EULAR) classification criteria [55]. Patients with knee OA were diagnosed based on the 2019 ACR recommendations [56]. Patients with traumatic meniscal injury (TMI) were diagnosed based on MRI descriptions of the Fischer grading criteria [57].

In the study, patients with TMI who did not exhibit radiographic signs of OA were incorporated into the control (CON) group. It is assumed that their knees were in a healthy state prior to the trauma and without any post-traumatic or OA-related abnormalities other than TMI. Patients diagnosed with OA were incorporated into the inflammation control (iCON) group. In every instance of RA assessed in the study, disease activity was evaluated using DAS28 (Disease activity scores in 28 joints) criterion, while disease progression was assessed using the Steinbrocker classification grade. Patients with RA were stratified into six distinct groups based on their disease activity and progression: RA activity with early-stage (aES RA), moderate-stage (aMS RA), and severe-stage (aSS RA) groups, RA remission with early-stage (rES RA), moderate-stage (rMS RA), and severe-stage (rSS RA) groups.

Disease onset was defined as the day of diagnosis, as documented in participant medical records. Only cases with a minimum of one year since diagnosis were considered for inclusion. The inclusion criteria were a primary diagnosis of RA, OA, or TMI. All patients included exhibited knee synovitis at the outset of the study and had not received intraarticular injections of steroids or chondroprotectors for at least three months prior to the study. The participants subsequently underwent either joint replacement or arthroscopic surgery for diagnostic and therapeutic purposes. The exclusion criteria for this study included a diagnosis of autoimmune diseases other than RA, specifically neuromyelitis optica, anti-myelin oligodendrocyte glycoprotein antibody-associated disorder, Lambert–Eaton myasthenic syndrome, multiple sclerosis, systemic lupus erythematosus, type 1 diabetes, hyperthyroidism, chronic thyroiditis, ulcerative colitis, Sjögren’s syndrome, ankylosing spondylitis, autoimmune hemolysis, idiopathic leukopenia, autoimmune glomerulonephritis, and idiopathic thrombocytopenic purpura. The rationale behind this decision was to eliminate potential confounding effects that these diverse autoimmune conditions could have on the study’s outcomes. Additional exclusion criteria comprised participation in intense or competitive activities for more than one hour per week in the past three months, use of glucocorticoids or biological disease-modifying anti-rheumatic drugs, presence of hemophilia, congenital patellar dysplasia, or other congenital knee deformities, acute and chronic infections, malignant diseases, severe lung, liver, kidney, or endocrinological disorders, and age younger than 18 or older than 80. Moreover, patients with OA or TMI were excluded if they had experienced an acute knee injury.

In total, 186 consecutive patients met the inclusion criteria. These individuals were screened according to the specific exclusion criteria within the outpatient department, resulting in 165 patients (88.71%) being categorized into the predefined groups. Each group of RA patients was divided by a 1:1 ratio according to their gender. Similarly, for each case in the CON group, we used a 1:6 ratio to organize RA patients according to gender. For the iCON group specifically, each case was divided by a 1:6 ratio based on the age (±5 years) and gender of the RA patients. This study ultimately involved 128 patients (68.82%) with 16 patients in each of the following groups: CON, iCON, aES RA, aMS RA, aSS RA, rES RA, rMS RA, and rSS RA groups. All 128 patients enrolled were tested for serum GRP78/Bip. Initially, serum GRP78/Bip levels were assessed in all 128 enrolled patients, as serum test results are more readily obtainable. Following this, patients were then selected for further studies based on the monitoring outcomes of serum GRP78/Bip. Despite continued enrollment over a three-year period, only 50 patients proceeded to surgical intervention. Subsequently, participants from each group were further divided into a 1:1 ratio according to gender. Within the same group, patients were subsequently screened based on their median serum levels of GRP78/Bip. Ultimately, 24 patients were involved, with 3 patients from each group included in the follow-up study.

### 2.3. Determination of RA Disease Activity

RA patients are distributed into two subgroups according to their DAS28 status: disease activity (DAS28 > 2.6) and disease remission (DAS28 ≤ 2.6) patients. DAS28 is calculated from four components, swollen joint count (SJC), tender joint count (TJC) of 28 joints, ESR (mm/h) as a marker of systemic inflammation, and global health assessment indicated by the visual analog scale (VAS).

### 2.4. Determination of RA Disease Progression

The Steinbrocker classification grade was used to organize the groups based on the severity of disease progression (Table 1). These stages can be divided into four categories based on clinical signs and radiographic changes. Specifically, patients with RA in Stage IV of the Steinbrocker classification were excluded from this study. In clinical observations, any two or more of these stages may occur simultaneously at different joints within the same patient, but usually the process advances to an approximately similar extent in multiple joints. Since it is not practical to evaluate the disease according to the stage of each affected joint separately, it is efficient to classify each case based on the condition of the specific joint under investigation, such as the knee joint in this particular study. It is crucial to recognize that the status of RA can be classified as either active or in remission at any given stage of disease progression.

### 2.5. Sample Collection and Preparation

Fasting blood samples were collected from participants in the morning using serum separator tubes under standard conditions. The samples were allowed to clot for 30 min at room temperature, followed by centrifugation at 1000× *g* for 10 min. The serum samples were then promptly aliquoted, kept on ice throughout the process, and stored at −80 °C.

SF samples were aspirated from the knee joint of patients prior to surgical procedures and extracted during capsular cutting in the surgical procedures. The samples were collected in heparinized containers, specifically containing heparin sodium at a concentration of 14 U/mL, to inhibit coagulation and avoid any potential cellular damage. After centrifugation at 1000× *g* for 20 min, the resulting cell-free SF was aliquoted and stored at −80 °C.

Synovium samples were collected during surgical procedures after ruling out infection. Samples from the CON group were obtained during arthroscopic synovectomy, while inflamed synovium samples for the iCON group were gathered during joint replacement surgeries. Inflamed synovium samples from aES RA and rES RA groups were also obtained during arthroscopic synovectomy, whereas samples for aMS RA, aSS RA, rMS RA, and rSS RA groups were gathered during joint replacement surgeries.

### 2.6. Enzyme-Linked Immunosorbent Assay (ELISA)

Levels of GRP78/Bip, tumor necrosis factor -α (TNF-α), and interleukin-10 (IL-10) in serum, SF, and synovium tissue homogenates were examined using ELISA kits (Jonin Biotech, Shanghai, China). All experimental protocols were strictly followed according to the manufacturer’s instructions.

### 2.7. Western Blot (WB)

Equal amounts of total protein from synovium lysates or SF samples were separated by 10% sodium dodecyl sulfate-polyacrylamide gel electrophoresis (SDS-PAGE) (1.5 M TRIS·Hcl, pH = 8.8, 30% acrylamide, 10% SDS, AP, TEMED) and then transferred onto nitrocellulose membranes (Millipore, Billerica, MA, USA). To prevent nonspecific binding, nonfat milk was used to block the membranes, after which the membranes were incubated and probed with rabbit polyclonal antibodies targeting GRP78/BiP (Proteintech Group, Wuhan, China) and Caspase-3 monoclonal antibodies (Abway Antibody Technology, Shanghai, China). Subsequently, the secondary antibody horseradish peroxidase (HRP)-conjugated anti-rabbit IgG (Servicebio Technology, Wuhan, China) was then applied. The purified protein’s molecular weight was determined by SDS-PAGE, and the band was excised for protein identification using a chemiluminescence detection system (Shenhua Science Technology, Hangzhou, China). Band density was analyzed with ImageJ 1.41 (National Institutes of Health, Bethesda, MD, USA).

### 2.8. Statistical Analysis

All data were presented as mean ± standard deviation ($\bar{x}$ ± s) or as median and interquartile range (IQR). The Shapiro–Wilk test was conducted to assess normal distribution of the data. In situations where the data does not adhere to a normal distribution, the Mann–Whitney U test was employed to assess the differences between two groups, while the Kruskal–Wallis H test was utilized for assessing differences among the three groups. Independent Student’s *t* test was used to compare the differences between two independent groups, and one-way analysis of variance (ANOVA) was applied to compare the differences among ≥3 independent groups. Spearman correlation coefficients were calculated to investigate the relationship between disease activity/severity status and the levels of GRP78/Bip, TNF-α, IL-10, or Caspase-3. The figure presented in this study was generated using Origin 8.5 software (OriginLab Corporation, Northampton, MA, USA). Statistical significance was defined as *p* < 0.05 for both differences and correlations.

## 3. Results

### 3.1. Serum GRP78/Bip Levels in Different States of RA

The serum GRP78/Bip levels were significantly higher in both RA remission and activity patients when contrasted with CON and iCON patients (*p* < 0.05) (Figure 1A,B, Table 2). Notably, the levels of serum GRP78/Bip were significantly elevated in RA activity patients compared to those in remission (*p* < 0.05) (Figure 1B). Additionally, serum GRP78/Bip levels in RA remission patients exhibited a notable increase associated with disease progression (*p* < 0.05) (Figure 1C). Spearman correlation analysis uncovered a significant positive correlation between serum GRP78/Bip levels and disease progression in RA remission patients (r = 0.388, *p* = 0.006). Furthermore, in RA activity patients, serum GRP78/Bip levels increased as disease progression escalated (*p* < 0.05) (Figure 1D). Spearman correlation analysis also demonstrated a significant positive correlation between serum GRP78/Bip levels and disease progression in RA activity patients (r = 0.437, *p* = 0.002).

### 3.2. The Differential Expressions of GRP78/BiP in Blood, SF, and Synovium Under Different Disease Activity and Progression of RA

As illustrated in Figure 2 and Table 3, a noteworthy increase in the levels of GRP78/Bip was observed in serum, SF, and synovium during both RA activity and remission statuses when compared to the CON and iCON groups (*p* < 0.05). Moreover, serum and synovium GRP78/Bip levels were significantly elevated during RA activity status compared to those during remission status (*p* < 0.05). Conversely, SF GRP78/Bip levels during RA activity status were lower than those observed during remission status (*p* < 0.05). The analysis of GRP78/Bip levels in serum, SF, and synovium revealed distinct patterns related to disease activity and progression of RA. During both disease activity and remission statuses of RA, patients in the severe-stage exhibited elevated serum GRP78/Bip levels compared to those in the moderate-stage, while the moderate-stage showed higher levels than the early-stage (*p* < 0.05). Similar patterns were observed in the synovium during RA activity status (*p* < 0.05); however, during RA remission status, the severe-stage demonstrated significantly higher levels of synovium GRP78/Bip than both the moderate- and early-stages (*p* < 0.05). Conversely, GRP78/Bip expression in SF was found to be elevated in the early- and moderate-stages during both RA activity and remission statuses, in comparison to the severe-stage (*p* < 0.05).

### 3.3. The Differential Expressions of GRP78/BiP and Caspase-3 Under Different Disease Activity and Progression of RA

The limited availability of pathological synovium tissue presents a challenge as the total protein content derived from individual samples is insufficient for comprehensive WB analysis. To address this issue, this study employed a strategy whereby tissue samples from three different patients in the same cohort were combined to form a sample pool. As illustrated in Figure 3 and Table 4, the levels of GRP78/Bip and Caspase-3 in synovium were notably elevated during both RA activity and remission statuses when compared to the CON and ICON groups (*p* < 0.05). Furthermore, GRP78/Bip and Caspase-3 levels in synovium when RA was active were greater than those observed during remission (*p* < 0.05). In contrast, the comparative evaluation of GRP78/Bip expression in SF indicated a tendency towards higher levels during the RA remission status; however, the difference when compared to RA activity status did not reach statistical significance (*p* > 0.05). The levels of GRP78/Bip in synovium demonstrated a clear pattern during RA remission status, with the highest levels observed in the severe-stage, followed by the moderate-stage, while the early-stage displayed the lowest levels (*p* < 0.05). During RA activity status, the severe-stage stood out as having significantly higher levels of GRP78/Bip when compared to both early-and moderate-stages cases (*p* < 0.05); however, no statistically significant difference was noted between the early- and moderate-stage cases (*p* > 0.05). Conversely, the levels of GRP78/Bip in SF were found to be higher in the early-stage compared to the moderate-stage, while the moderate-stage exhibited greater levels than the severe-stage during RA activity status (*p* < 0.05). During the remission status of RA, levels of SF GRP78/Bip were significantly elevated in both the early- and moderate-stages relative to the severe-stage (*p* < 0.05), with no notable difference detected between the early- and moderate-stages (*p* > 0.05). Additionally, Caspase-3 levels in synovium were persistently elevated during both activity and remission of RA for cases classified as moderate- and severe-stages when compared to early-stage cases (*p* < 0.05). Notably, no noteworthy difference in Caspase-3 expression was observed between moderate- and severe-stages cases (*p* > 0.05).

### 3.4. The Differential Expressions of Inflammatory Factor TNF-α and Anti-Inflammatory Factor IL-10 Under Different Disease Activity and Progression of RA

As illustrated in Figure 4 and Table 5 and Table 6, TNF-α and IL-10 levels in serum, SF, and synovium were significantly elevated during both RA activity and remission statuses when compared to the CON and iCON groups (*p* < 0.05). Furthermore, the levels of TNF-α and IL-10 in serum and synovium exhibited a notable increase during disease activity compared to those in disease remission (*p* < 0.05). Notably, TNF-α levels in SF were higher during RA activity compared to remission (*p* < 0.05), while IL-10 levels displayed a contrasting trend, being lower during RA activity (*p* < 0.05). In serum, the levels of TNF-α and IL-10 consistently rose as the disease progressed from early- to moderate- and then to the severe-stages during RA activity (*p* < 0.05). During the remission of RA, TNF-α and IL-10 levels were higher in the severe-stage when compared to both the early- and moderate-stages, while there was no significant difference between early- and moderate-stages (*p* > 0.05). Concerning SF, TNF-α levels progressively increased from early- to moderate- and then to severe-stages during RA remission status, with significantly higher levels in both moderate- and severe-stages compared to the early-stage during activity status (*p* < 0.05). IL-10 levels also showed an increase in moderate- and severe-stages compared to early-stage during RA activity (*p* < 0.05), but there were no significant variations among different stages during remission status (*p* > 0.05). Regarding synovium, TNF-α and IL-10 levels during RA remission status were found to be highest in the severe-stage, then in moderate-stage, and lowest in early-stage (*p* < 0.05). In the context of RA activity status, the levels in both moderate- and severe-stages showed no significant difference (*p* > 0.05); however, both were significantly greater than those observed in early-stage (*p* < 0.05).

## 4. Discussion

The current recommended standard for RA diagnosis was the 2010 ACR/EULAR classification criteria [55]. These criteria include clinical assessments such as joint counts, acute phase reactants (APR), and crucially, the identification of rheumatoid factor (RF) and anticitrullinated protein/peptide antibodies (ACPAs). RF and ACPAs stood out as the most frequently used biomarkers in clinical settings for guiding the diagnosis and prognosis of RA. The sensitivity of RF in diagnosing RA ranges from 30% to 70% in early cases and increases to 80–85% in progressive cases, but its specificity is ~40%, as it could also be present in patients with other diseases and even in healthy individuals [25,45]. In contrast, ACPAs exhibited a moderate sensitivity of 60~80% and a high specificity of 95~98% [25]. Despite their widespread use, both RF and ACPA, along with APR, have proven to be inadequate in satisfactorily responding to the high heterogeneity of RA [59,60]. Clinical practice still faced challenges with misdiagnosis and missed diagnoses, which could result in unwarranted initiation of DMARD therapy [59,60]. Therefore, issues arose on how comprehensive the criteria should be and whether it should be updated and adapted to findings from the past two decades that might increase both its specificity and sensitivity [59,60]. Additionally, the added value of other anti-modified protein antibodies or biomarkers to the increased sensitivity and/or specificity of the criteria remains a topic of debate [59,60].

Biomarkers are defined as interaction parameters that provide objective information on measurable changes in physiology, biochemistry, or morphology, which are evaluable at the molecular, biochemical, or cellular level [61]. These biomarkers serve as indicators of functional biological processes, pathogenic states, or responses to medical interventions [62]. Ideal biomarkers should be able to provide diagnostic, prognostic, and therapeutic information [25]. They should be obtainable from a patient’s clinical data and possess specific chemical-analytical characteristics [25]. These characteristics include high specificity, where the measurement of a biomarker must be specific to a disease; specimen collection should be minimally invasive, with saliva being preferred over urine and urine over blood; representativeness, meaning levels of biomarkers in the sample should be representative of levels in the organism; and stability, where the kinetics must be known [25]. The identification of RF and ACPAs as biomarkers has created new opportunities for diagnosing RA and predicting its progression. However, there is currently no ideal biomarker that fully meets all desired characteristics. Consequently, the search for new biomarkers with genuine clinical utility remains a major topic of interest in RA research.

Recent advancements in laboratory techniques have facilitated the discovery of additional novel autoantibodies, particularly those targeting post-translationally modified proteins, thereby expanding the range of potential biomarkers for RA. One promising newly identified autoimmune biomarker for RA is GRP78/BiP. This biomarker is noteworthy due to its significance in disease processes, as well as its high specificity, remarkable stability, and pronounced expression in saliva [41], blood [36,42], SF [42,53], and synovium [23,36]. The identification of GRP78/BiP-specific autoantibodies has emerged as a novel diagnostic tool, potentially serving as both a preclinical indicator [23,42] and an improved diagnostic biomarker [25] for RA. Recent integrative studies evaluating the diagnostic value of GRP78/BiP and anti-GRP78/BiP antibodies in saliva, blood, SF, and synovium for RA has demonstrated a high pooled specificity of 0.92 and a moderate pooled sensitivity of 0.67 [60]. The overall diagnostic odds ratio (DOR) was notably high at 23.73, while the pooled positive likelihood ratio (LR+) was 8, indicating that individuals diagnosed with RA are eight times more likely to test positive for biomarkers associated with GRP78/BiP compared to those without the condition. Consequently, GRP78/BiP-specific autoantibodies could serve as a valuable supplement to current diagnostic strategies, thereby enhancing diagnostic precision and contribute to more informed clinical decision-making [23,25,36,41,42,43,53,60].

The existing research on the potential clinical biomarker GRP78/Bip in RA has primarily been limited to isolated examinations of either synovium [23,36,37,38,39,40], blood [14,36,39,40,42,43,44,45,46,47,48,49,50,51], or SF [42,44,52,53,54]. However, no studies have comprehensively examined GRP78/Bip across blood, SF, and synovium collectively. Due to the insufficiency of relevant studies, the current research has not conducted stratified analyses on specific subgroups such as blood, SF, and synovium. Understanding the pathogenesis and progression of RA requires an analysis of both systemic and local immune responses, which are intricately linked. Investigating the changes and interconnections of GRP78/Bip in different intracorporeal environments, including blood, SF, and synovium, is essential for gaining a comprehensive understanding of its role as a potential biomarker in RA. By adopting a holistic approach to study GRP78/Bip, researchers may uncover valuable insights into its significance and implications for RA treatment and management. Furthermore, RA is a complex systemic disease characterized by gradual advancement and deterioration over time. The development of RA is closely associated with the various stages of disease progression and the fluctuating statuses of disease activity [16,63,64]. Despite this understanding, there is a notable lack of comprehensive research into the differential expressions of GRP78/Bip levels in relation to the different statuses and stages of RA. Further investigation into these factors is necessary to gain a better understanding of the role of GRP78/Bip in the pathogenesis of RA. Hence, this study delves into the differential expressions of GRP78/Bip in blood, SF, and synovium under different conditions, statuses, and stages of RA. By examining these different biological contexts, the research aims to provide insights into how the expression of GRP78/Bip may change in response to the statuses and stages of RA. Furthermore, this research endeavors to explore the responsibilities, properties, and associations of the intracellular and extracellular GRP78/Bip with the disease activity and progression in RA.

The results of this study revealed that GRP78/Bip was consistently present in both the intracellular environment (synovium) and the extracellular environment (cell-free blood and SF) in RA, with fluctuations in expression levels closely linked to the disease activity and progression of RA. These findings underscore the significance of GRP78/Bip in the pathogenesis of RA, suggesting that its expression levels could serve as valuable indicators of disease activity and progression in RA. The disease activity statuses and progression stages of RA were found to align with the expressions of GRP78/Bip in blood and synovium, but differed significantly from those observed in SF. This highlights the critical role of GRP78/Bip in the disease activity and progression of RA, noting that its function may vary depending on the specific intracorporeal environment, as well as the statuses and stages of RA. To gain a deeper understanding of the significant discrepancies observed, a thorough and comprehensive analysis was undertaken. This investigation not only examined the results of our own research but also integrated valuable insights derived from prior research in the field [14,21,22,27,28,30,65,66,67,68,69,70,71]. By leveraging existing literature and earlier findings, the analysis aimed to illuminate the fundamental reasons behind these notable differences, thereby contributing to a more nuanced understanding of the issue in RA (Figure 5).

The investigation into the function of GRP78/Bip in the synovium during disease activity and progression of RA was performed. Upon the onset of RA, elevated levels of intracellular GRP78/Bip are generated in response to stress signals, aimed at disturbances in the ER. This process facilitates the re-establishing ER functionality, mitigates cellular damage, and supports cell viability. As ER stress persists and disease advances, the body increasingly activates stress response mechanisms in the stressed cells. This process aims to prevent irreversible damage and apoptosis of these stressed cells through the expression of intracellular GRP78/Bip. Consequently, the expression of GRP78/Bip in synovium increases gradually with the severity of the disease. Specifically, in this study, the levels of intracellular GRP78/Bip expression in synovium were found to be highest at severe-stage, followed by moderate-stage, and lowest in early-stage. In the context of RA, disease activity resembles an acute exacerbation, characterized by heightened apoptosis and intensified inflammatory reactions in synovium. Notably, elevated levels of caspase-3 expression were observed during disease activity status in this study, as opposed to remission status, with higher expression levels in severe- and moderate-stages compared to early-stage. This RA activity status also witnesses an aggravated inflammatory response and a rapid increase in the expression and synthesis of intracellular GRP78/Bip. This upregulation of intracellular GRP78/Bip serves as a protective mechanism against further progression of the inflammatory disease. Conversely, during RA remission status, the inflammatory response is alleviated, leading to a decrease in intracellular GRP78/Bip expression, but still remains relatively high compared to healthy and disease control individuals. The results of this study demonstrated that the expression levels of GRP78/Bip in synovium were consistently greater during disease activity status compared to those in remission status.

The investigation also examined the function of GRP78/Bip in the SF. In the context of RA onset, GRP78/Bip experiences a shift in location and function within the cell. Normally residing in the ER, GRP78/Bip can avoid ER “reattachment” by binding to unfolded proteins. However, during the development of RA, GRP78/Bip can translocate to the cytoplasm and cell membrane along with these unfolded proteins. As ER stress disrupts ER structure, GRP78/Bip is released into the extracellular space. This results in the loss of its anti-apoptotic effect within the cell but leads to its involvement in the immune response and anti-inflammatory response outside the cell as extracellular GRP78/Bip and autoantigens. As the disease progresses and worsens, some intracellular GRP78/Bip have already depleted due to apoptosis before metastasizing to the extracellular space, further decreasing the extracellular expression. However, the disruption of homeostatic function and the increase in apoptosis in stressed cells will in turn exacerbate protein misfolding and accumulation and further increase the intensity of ER stress response, thus leading to a vicious cycle and accelerating cell dysfunction and lesion progression. As ER stress persists and disease advances, the ability to maintain homeostasis diminishes, resulting in an increased demand for intracellular GRP78/Bip. Consequently, there is a reduced release of intracellular GRP78/Bip into the extracellular space. When ER stress-induced apoptosis exceeds its protective capabilities and causes irreversible damage, apoptotic signals are activated to remove stressed cells if normal function cannot be restored. This helps maintain a delicate balance between cell survival and apoptosis. In such situations, some intracellular GRP78/Bip may have been lost due to apoptosis prior to metastasizing to the extracellular space, further reducing its expression outside the cell. This disruption of homeostatic function, coupled with an increase in apoptosis in stressed cells, will exacerbate protein misfolding and accumulation, intensify ER stress response, creating a harmful cycle that accelerates cell dysfunction and disease progression. Therefore, the aforementioned scenario presents a negative correlation between extracellular GRP78/Bip levels and severity of RA progression. This study further confirmed the presence of extracellular GRP78/Bip in SF of patients with RA at different disease progression stages, with levels decreasing from early- to moderate- and then to severe-stages. Additionally, during RA activity, intracellular GRP78/Bip was presented to be significantly elevated, resulting in a relatively lower extracellular GRP78/Bip level in SF. Notably, this study demonstrated that the extracellular levels of GRP78/Bip in SF exhibited a pattern where disease remission status was associated with higher levels than those observed during disease activity status in RA patients.

Additionally, the function of GRP78/Bip in the bloodstream was examined. GRP78/Bip serves as a reactive protective mechanism in cells, emphasizing its essential role in maintaining cellular function and viability. The presence of both intracellular and extracellular GRP78/Bip has been shown to work together to exhibit synergistic effects, further highlighting the significance of this protein in cellular homeostasis. As RA develops, there is a gradual increase in the body’s requirement for GRP78/Bip to cope with the ongoing inflammatory processes. In response to this heightened demand, the immune system is activated to increase the production of GRP78/Bip at the affected area. This results in elevated levels of GRP78/Bip in the bloodstream, reflecting the body’s efforts to counteract the inflammatory response associated with RA. However, achieving remission in RA is contingent upon striking a delicate balance between intracellular and extracellular GRP78/Bip expression at the affected site. This equilibrium is essential for resolving inflammation and restoring normal cellular function. Ultimately, the body’s demand for circulating GRP78/Bip must be maintained at a relatively balanced state to facilitate recovery from RA and promote overall well-being. Consequently, the findings of this research revealed distinct patterns in GRP78/Bip expression in the blood of RA patients: disease activity was found to exceed that of disease remission, while severe-stage disease surpassed moderate-stage, and moderate-stage surpassed early-stage.

Furthermore, a study was performed to evaluate the fluctuations in both inflammatory and anti-inflammatory factors present within blood, SF, and synovium. The results indicated that the levels of inflammatory and anti-inflammatory cytokines in individuals with RA varied depending on their disease activity statues and progression stages. As ER stress persists and RA lesions aggravate gradually, the inflammatory response intensifies both locally and systemically. This escalation results in elevated expression of both inflammatory and anti-inflammatory factors in various areas such as blood, SF, and synovium. Notably, the levels of TNF-α and IL-10 in these regions exhibited a progressive increase correlating with the progression of RA, with the highest levels observed in severe-stage, followed by moderate-stage and then early-stage. This pattern indicates that as RA progresses, the inflammatory response intensifies at various levels within the body. During RA activity status, the disease reaches a phase of acute exacerbation. In this condition, both local and systemic inflammatory responses are significantly more severe in comparison to the remission status. This heightened inflammatory activity leads to a significant increase in the secretion of inflammatory factors by the body. In this study, TNF-α levels in blood, SF, and synovium were notably higher during disease activity as opposed to remission. In the acute exacerbation state of disease activity, the local inflammatory response is notably more intense than the anti-inflammatory response. To counteract this, the body mobilizes both local and systemic cellular immune systems to alleviate the inflammation. This process results in the secretion of numerous anti-inflammatory factors from synovium and blood, which are rich in cellular components. It was observed that the level of anti-inflammatory factor IL-10 in synovium and blood was higher during disease activity when compared to remission. However, the production of IL-10 in SF, which contains fewer cells, was limited. During disease remission status, the body maintains a dynamic balance between local and systemic inflammatory and immune responses. Although the local inflammatory response remains high, there is a relatively balanced interplay between inflammatory and anti-inflammatory factors. This equilibrium is achieved by enhancing local immune responses, leading to increased production of anti-inflammatory factors. For example, in this study, it was observed that the levels of IL-10 in SF were elevated during RA remission status when contrasted with activity status.

This research indicated that patients with RA experience a shift in the balance between inflammatory and anti-inflammatory factors under varying environmental conditions. Specifically, TNF-α was identified as the primary factor present in SF, exhibiting higher levels in the bloodstream compared to IL-10. However, only minor discrepancies were observed in the levels of TNF-α and IL-10 within the synovium. Simultaneously, this study demonstrated that changes in Caspase-3 levels correlated with alterations in GRP78/Bip within the inflamed synovium. Furthermore, variations in IL-10 levels in blood, SF, and inflamed synovium closely mirrored the patterns observed in GRP78/Bip, thereby supporting the anti-inflammatory properties of extracellular GRP78/Bip. These findings suggest that GRP78/Bip may play a significant role in modulating immune responses, apoptotic processes, and the reduction in inflammation in both local and systemic environments.

The terminological issues surrounding the pathological processes of RA are significant. As we explored these concepts, it became apparent that considerable inconsistencies exist in the terminology employed within the field [1,4]. In response to this challenge, we conducted this comprehensive study aimed at standardizing the terminology related to these factors. Our objective was to establish a clear and consistent framework that would serve as a valuable reference for scholars and researchers in future investigations. By addressing these terminological discrepancies, we hope to facilitate more effective communication and understanding within the academic community. We propose that the onset and development of RA can be understood through three crucial phases: disease occurrence, disease activity, and disease progression, which correspond to three states in RA: the conditions of RA, statuses of RA, and stages of RA. The conditions of RA are associated with the likelihood of disease occurrence. The statuses of RA pertain to disease activity or remission that patients may experience throughout their illness, while the stages of RA are linked to the disease’s progression over time. Understanding RA in this manner allows for a more comprehensive approach to treatment strategies and patient management.

Several limitations were present in this study, with the most significant being the small sample size and monocentric design. The limited availability of pathological synovium samples constrains the variety of analytical methods that can be employed, including immunohistochemistry and immunofluorescence. This scarcity further restricts the comprehensiveness of research findings, potentially impacting the overall conclusions drawn from the data. Further investigation into apoptosis and cleaved caspases is warranted, as indicated by our findings. Participants were recruited exclusively from a single hospital containing exclusively Chinese racial groups, which may have restricted their representativeness and the generalizability of the results to a broader population. Future research should incorporate a larger participant pool to introduce more significant findings and examine subjects from diverse racial and geographical backgrounds. Another significant limitation pertains to the absence of investigation into the presence of GRP78/Bip in healthy individuals due to ethical constraints, thus weakening the ability to establish a clear relationship between GRP78/Bip and the activity and progression of RA. Future research should aim to address these limitations by ensuring comprehensive data collection from both healthy individuals and RA patients to provide a more robust understanding of the role of GRP78/Bip in the disease process. Additionally, this was a cross-sectional study, thus the causalities between GRP78/Bip and RA remain unverified. Further validation is needed through more extensive prospective studies, such as longitudinal or cohort studies. Furthermore, the normal levels of GRP78/Bip in serum, SF, and synovium have not been definitively established, nor have we fully understood the significance of these levels. Various factors including age, sex, diet, ethnicity, physical activity, medication use, the method of estimating GRP78/Bip, and sample storage conditions all have the potential to influence GRP78/Bip levels. Further research is needed to establish a baseline for normal GRP78/Bip levels, understand its implications, and account for potential confounding variables that could affect measurements. Last, exploring how interventions specifically targeted at intracellular or extracellular GRP78/Bip impact the activity and progression of RA in animal models or cell-based research was not within the scope of this study. Overall, while this study did not directly investigate the effects of interventions targeting GRP78/Bip on RA, it highlights the need for future research in this area to better understand the role of GRP78/Bip in the pathophysiology of RA and to potentially identify novel treatment strategies for the disease.

## 5. Conclusions

The GRP78/Bip expression levels in blood, SF, and synovium of patients with RA are significantly higher than those observed in OA and TMI. Furthermore, the expression levels of GRP78/Bip in blood and synovium demonstrate a positive correlation with disease activity and progression of RA, whereas GRP78/Bip in SF demonstrate a negative correlation with disease activity and progression. These differential expressions of GRP78/Bip in blood, SF, and synovium are associated with disease activity and progression in RA. While this study was the first to explore the differential expressions and significance of intracellular and extracellular GRP78/Bip in relation to disease activity and progression of RA, and points to the significance of subject-specific changes in GRP78/Bip in serum, SF, and synovium, additional molecular cell research, animal or in vitro experiments, and clinical studies are necessary to evaluate the effects of GRP78/Bip so as to provide further comprehensive evidence and guidance for its diagnostic value and clinical application in RA.

## Figures and Tables

**Figure 1 bioengineering-12-00058-f001:**
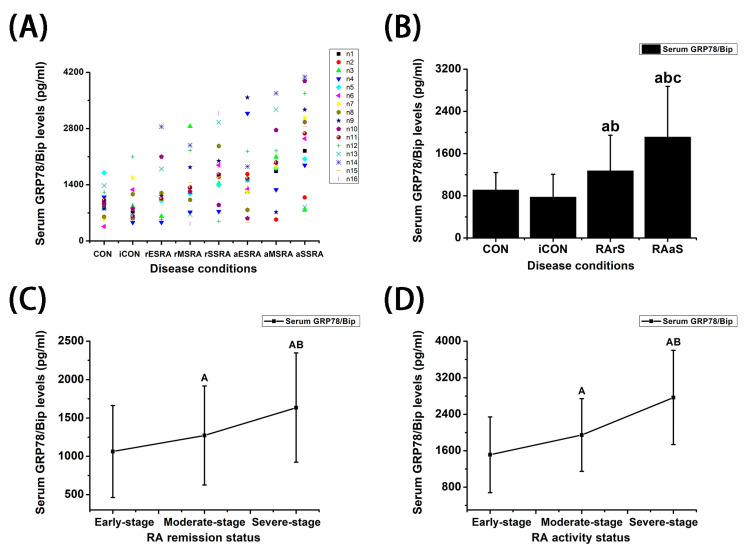
The differential expressions of serum GRP78/BiP in different states of RA. Serum GRP78/BiP under different conditions (*n* = 16) (**A**), statuses (*n* = 16, 16, 48, 48) (**B**), and stages (*n* = 32) [(**C**,**D**)] of RA. CON: Patients with traumatic meniscal injury (*n* = 16). iCON: Patients with osteoarthritis. RA activity patients with early-stage (aES RA), moderate-stage (aMS RA), and severe-stage (aSS RA). RA remission patients with early-stage (rES RA), moderate-stage (rMS RA), and severe-stage (rSS RA). RArS: disease remission status of RA. RAaS: disease activity status of RA. Mann–Whitney U test was employed to assess the differences between two groups. a: *p* < 0.05 versus CON. b: *p* < 0.05 versus iCON. c: *p* < 0.05 versus RArS. A: *p* < 0.05 versus early-stage. B: *p* < 0.05 versus moderate-stage.

**Figure 2 bioengineering-12-00058-f002:**
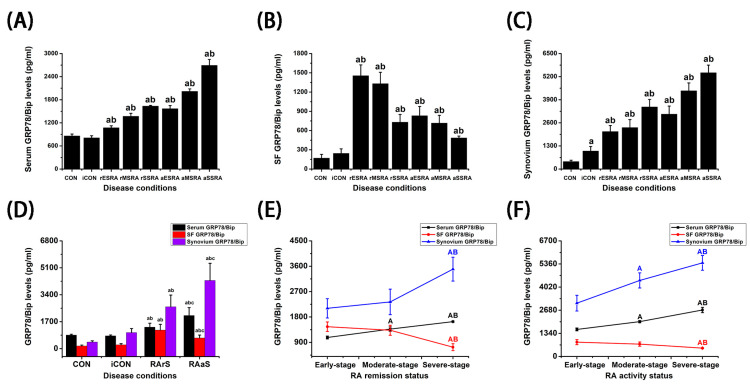
The differential expressions of GRP78/BiP in different states of RA. GRP78/BiP levels in serum, SF, and synovium under different conditions (*n* = 3) [(**A**–**C**)], statuses (*n* = 3, 3, 9, 9) (**D**), and stages (*n* = 6) [(**E**,**F**)] of RA. SF: synovial fluid. CON: Patients with traumatic meniscal injury. iCON: Patients with osteoarthritis. RA activity patients with early-stage (aES RA), moderate-stage (aMS RA), and severe-stage (aSS RA). RA remission patients with early-stage (rES RA), moderate-stage (rMS RA), and severe-stage (rSS RA). RArS: RA remission status. RAaS: RA activity status. Independent Student’s *t* test was used to compare the differences between two independent groups. a: *p* < 0.05 versus CON. b: *p* < 0.05 versus iCON. c: *p* < 0.05 versus RArS. A: *p* < 0.05 versus early-stage. B: *p* < 0.05 versus moderate-stage.

**Figure 3 bioengineering-12-00058-f003:**
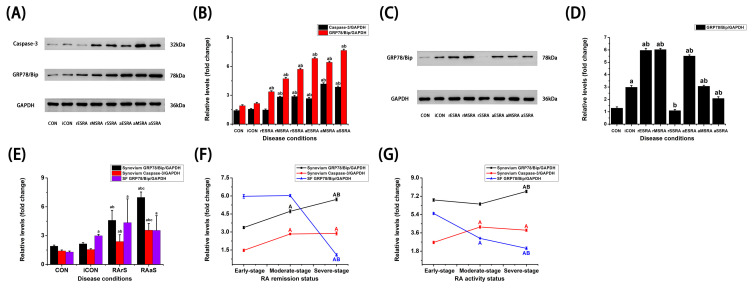
The expressions of GRP78/BiP and Caspase-3 in different states of RA. Western blot analysis of GRP78/BiP and Caspase-3 in synovium [(**A**,**B**)], GRP78/BiP in SF [(**C**,**D**)]. GRP78/BiP and Caspase-3 levels under different statuses (**E**), and stages [(**F**,**G**)] of RA. SF: synovial fluid. CON: Patients with traumatic meniscal injury. iCON: Patients with osteoarthritis. RA activity patients with early-stage (aES RA), moderate-stage (aMS RA), and severe-stage (aSS RA). RA remission patients with early-stage (rES RA), moderate-stage (rMS RA), and severe-stage (rSS RA). RArS: RA remission status. RAaS: RA activity status. Independent Student’s *t* test was used to compare the differences between two independent groups. a: *p* < 0.05 versus CON. b: *p* < 0.05 versus iCON. c: *p* < 0.05 versus RArS. A: *p* < 0.05 versus early-stage. B: *p* < 0.05 versus moderate-stage.

**Figure 4 bioengineering-12-00058-f004:**
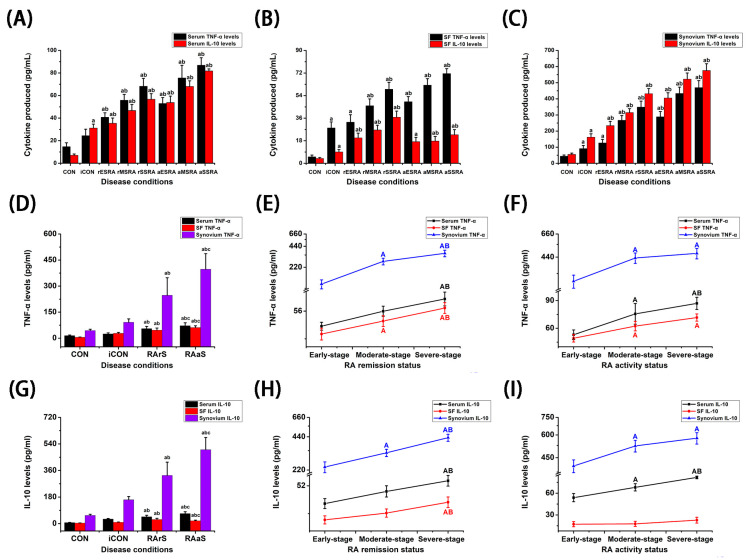
The expressions of TNF-α and IL-10 in different states of RA. TNF-α and IL-10 levels in serum (**A**), SF (**B**), and synovium (**C**) under different disease conditions of RA (*n* = 3). TNF-α levels under different statuses (*n* = 3, 3, 9, 9) (**D**), and stages (*n* = 6) [(**E**,**F**)] of RA. IL-10 levels under different statuses (*n* = 3, 3, 9, 9) (**G**), and stages (*n* = 6) [(**H**,**I**)] of RA. SF: synovial fluid. CON: Patients with traumatic meniscal injury. iCON: Patients with osteoarthritis. RA activity patients with early-stage (aES RA), moderate-stage (aMS RA), and severe-stage (aSS RA). RA remission patients with early-stage (rES RA), moderate-stage (rMS RA), and severe-stage (rSS RA). RArS: RA remission status. RAaS: RA activity status. Independent Student’s *t* test was used to compare the differences between two independent groups. a: *p* < 0.05 versus CON. b: *p* < 0.05 versus iCON. c: *p* < 0.05 versus RArS. A: *p* < 0.05 versus early-stage. B: *p* < 0.05 versus moderate-stage.

**Figure 5 bioengineering-12-00058-f005:**
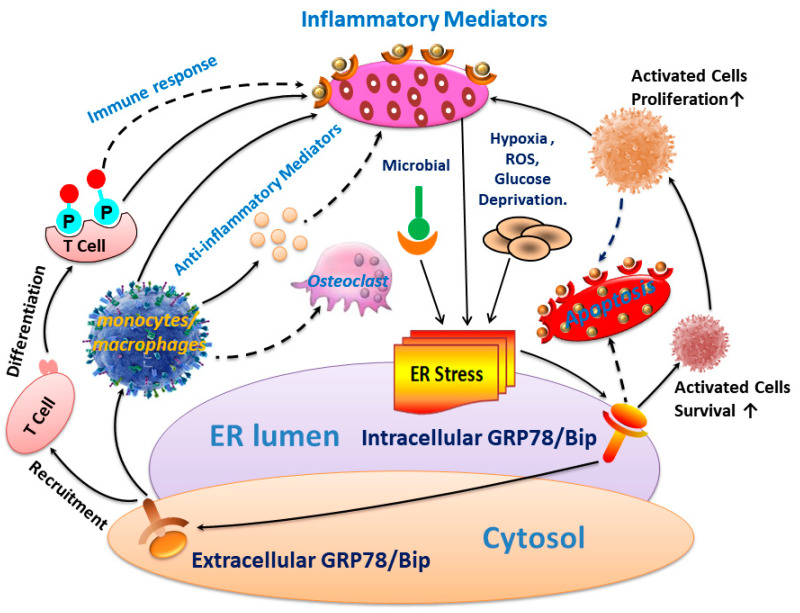
The intracellular and extracellular GRP78/BiP is a master regulator of apoptosis and inflammation, obtaining help from GRP78/BiP in RA.

**Table 1 bioengineering-12-00058-t001:** Classification of RA disease progression [58].

Stages	Items
Stage I (Early-stage)	* 1. No destructive changes roentgenologically. 2. Roentenologic evidence of osteoporosis may be present.
Stage II (Moderate-stage)	* 1 Roentgenologic evidence of osteoporosis, with or without slight subchondral bone destruction; slight cartilage destruction may be present. * 2. No joint deformities, although limitation of joint mobility may be present. 3. Adjacent muscle atrophy. 4. Extra-articular soft tissue lesions, such as nodules and tenovaginitis, may be present.
Stage III (Severe-stage)	* 1. Roentgenologic evidence of cartilage and bone destruction, in addition to osteoporosis. * 2. Joint deformity, such as subluxation, ulnar deviation or hyperextension, without fibrous or bony ankylosis. 3. Extensive muscle atrophy. 4. Extra-articular soft tissue lesions, such as nodules and tenovaginitis, may be present.
Stage IV (Terminal-stage)	* 1. Fibrous or bony ankylosis. 2. Criteria of stage III.

* Indicates criteria required to be present.

**Table 2 bioengineering-12-00058-t002:** Serum GRP78/Bip in different states of RA (pg/mL).

States	Cases	Serum GRP78/Bip
Conditions of RA		
CON	16	909.62 (652.43, 1072.09)
iCON	16	775.36 (592.86, 1101.04)
RA		1605.54 (1081.84, 2239.68) ^ab^
*χ*^2^-value		27.132
*p*-value		<0.001
Statuses of RA		
Disease remission	48	1272.28 (1024.13, 1826.40) ^ab^
Disease activity	48	1913.08 (1298.23, 2831.98) ^abc^
*Z*-value		−3.411
*p*-value		0.001
Stages of RA (disease remission status)		
early-stage	16	1062.73 (1014.98, 1141.36)
moderate-stage	16	1272.27 (1137.50, 1554.30) ^A^
severe-stage	16	1634.72 (1419.49, 1914.95) ^AB^
*χ*^2^-value		7.066
*p*-value		0.029
Stages of RA (disease activity status)		
early-stage	16	1511.95 (1225.85, 1710.79)
moderate-stage	16	1945.29 (1795.40, 2128.20) ^A^
severe-stage	16	2766.86 (2002.97, 3117.49) ^AB^
*χ*^2^-value		8.956
*p*-value		0.011

^a^: *p* < 0.05 versus CON. ^b^: *p* < 0.05 versus iCON. ^c^: *p* < 0.05 versus disease remission. ^A^: *p* < 0.05 versus early-stage. ^B^: *p* < 0.05 versus moderate-stage.

**Table 3 bioengineering-12-00058-t003:** Serum, SF, and synovium GRP78/BiP in different states of RA (pg/mL).

States	Cases	Serum GRP78/Bip	SF GRP78/Bip	Synovium GRP78/Bip
Conditions of RA				
CON	3	859.55 ± 50.30	170.74 ± 57.67	419.13 ± 78.82
iCON	3	810.93 ± 54.35	244.86 ± 70.11	1017.88 ± 249.58
RA	18	1726.85 ± 537.09 ^ab^	924.60 ± 376.90 ^ab^	3476.43 ± 1239.18 ^ab^
*F*-value		7.650	10.021	13.911
*p*-value		0.003	0.001	<0.001
Statuses of RA				
Disease remission	9	1360.28 ± 247.75 ^ab^	1171.94 ± 362.71 ^ab^	2646.48 ± 735.74 ^ab^
Disease activity	9	2093.41 ± 499.27 ^abc^	677.27 ± 180.72 ^abc^	4306.37 ± 1082.57 ^abc^
*F*-value		15.572	13.411	14.473
*p*-value		0.001	0.002	0.002
Stages of RA (disease remission status)				
early-stage	6	1073.55 ± 49.98	1455.30 ± 168.45	2107.61 ± 341.93
moderate-stage	6	1372.14 ± 76.17 ^A^	1330.56 ± 178.23	2335.13 ± 452.87
severe-stage	6	1635.14 ± 19.23 ^AB^	729.96 ± 121.99 ^AB^	3496.72 ± 421.79 ^AB^
*F*-value		81.958	18.042	9.994
*p*-value		<0.001	0.012	0.003
Stages of RA (disease activity status)				
early-stage	6	1569.17 ± 81.94	831.60 ± 144.65	3089.56 ± 456.12
moderate-stage	6	2016.92 ± 65.42 ^A^	716.10 ± 122.01	4406.81 ± 438.31 ^A^
severe-stage	6	2694.15 ± 154.10 ^AB^	484.11 ± 29.61 ^AB^	5422.73 ± 426.90 ^AB^
*F*-value		83.105	7.683	21.147
*p*-value		<0.001	0.022	0.002

^a^: *p* < 0.05 versus CON. ^b^: *p* < 0.05 versus iCON. ^c^: *p* < 0.05 versus disease remission. ^A^: *p* < 0.05 versus early-stage. ^B^: *p* < 0.05 versus moderate-stage.

**Table 4 bioengineering-12-00058-t004:** Relative levels of GRP78/BiP and Caspase-3 in different states of RA (fold change).

States	Synovium		SF
	GRP78/Bip	Caspase-3	GRP78/Bip
Conditions of RA			
CON	1.91 ± 0.11	1.42 ± 0.09	1.30 ± 0.10
iCON	2.15 ± 0.12	1.56 ± 0.08	2.99 ± 0.14 ^a^
RA	5.79 ± 1.46 ^ab^	2.98 ± 0.91 ^ab^	3.96 ± 2.03 ^a^
*F*-value	18.376	7.512	2.867
*p*-value	<0.001	0.003	0.079
Statuses of RA			
Disease remission	4.60 ± 1.02 ^ab^	2.39 ± 0.70 ^ab^	4.36 ± 2.46 ^a^
Disease activity	6.98 ± 0.56 ^abc^	3.57 ± 0.70 ^abc^	3.55 ± 1.53 ^a^
*F*-value	54.472	13.788	2.197
*p*-value	<0.001	<0.001	0.120
Stages of RA (disease remission status)			
early-stage	3.37 ± 0.10	1.47 ± 0.10	5.97 ± 0.16
moderate-stage	4.72 ± 0.13 ^A^	2.83 ± 0.07 ^A^	6.03 ± 0.10
severe-stage	5.71 ± 0.12 ^AB^	2.88 ± 0.13 ^A^	1.09 ± 0.10 ^AB^
*F*-value	305.340	187.301	1593.076
*p*-value	<0.001	<0.001	<0.001
Stages of RA (disease activity status)			
early-stage	6.83 ± 0.13	2.66 ± 0.12	5.51 ± 0.09
moderate-stage	6.43 ± 0.12	4.18 ± 0.12 ^A^	3.06 ± 0.05 ^A^
severe-stage	7.67 ± 0.13 ^AB^	3.86 ± 0.11 ^A^	2.08 ± 0.13 ^AB^
*F*-value	76.103	145.204	853.964
*p*-value	<0.001	<0.001	<0.001

^a^: *p* < 0.05 versus CON. ^b^: *p* < 0.05 versus iCON. ^c^: *p* < 0.05 versus disease remission. ^A^: *p* < 0.05 versus early-stage. ^B^: *p* < 0.05 versus moderate-stage.

**Table 5 bioengineering-12-00058-t005:** TNF-α levels in different states of RA (pg/mL).

States	Cases	Serum TNF-α	SF TNF-α	Synovium TNF-α
Conditions of RA				
CON	3	14.70 ± 3.44	5.25 ± 1.32	44.20 ± 7.28
iCON	3	24.47 ± 5.76	28.22 ± 4.79 ^a^	91.50 ± 19.25
RA		63.40 ± 16.76 ^ab^	53.49± 13.56 ^ab^	322.24 ± 120.37 ^ab^
*F*-value		18.953	22.719	12.515
*p*-value		<0.001	<0.001	<0.001
Statuses of RA				
Disease remission	9	54.97 ± 12.83 ^ab^	45.95 ± 12.38 ^ab^	247.42 ± 100.76 ^ab^
Disease activity	9	71.83 ± 16.51 ^abc^	61.03 ± 10.47 ^abc^	397.06 ± 89.67 ^abc^
*F*-value		5.853	7.784	11.077
*p*-value		0.028	0.013	0.004
Stages of RA (disease remission status)				
early-stage	6	40.77 ± 3.99	32.81 ± 5.95	126.47 ± 21.65
moderate-stage	6	55.90 ± 5.10	45.94 ± 5.40 ^A^	267.13 ± 29.21 ^A^
severe-stage	6	68.25 ± 6.97 ^AB^	59.11 ± 5.43 ^AB^	348.67 ± 37.21 ^AB^
*F*-value		13.688	13.889	39.572
*p*-value		0.006	0.006	<0.001
Stages of RA (disease activity status)				
early-stage	6	53.00 ± 5.22	49.17 ± 4.04	288.17 ± 32.90
moderate-stage	6	75.60 ± 11.17 ^A^	62.35 ± 5.14 ^A^	433.33 ± 39.33 ^A^
severe-stage	6	86.90 ± 6.63 ^AB^	71.58 ± 3.90 ^A^	469.67 ± 43.16 ^A^
*F*-value		18.878	19.675	18.474
*p*-value		0.003	0.002	<0.001

^a^: *p* < 0.05 versus CON. ^b^: *p* < 0.05 versus iCON. ^c^: *p* < 0.05 versus disease remission. ^A^: *p* < 0.05 versus early-stage. ^B^: *p* < 0.05 versus moderate-stage.

**Table 6 bioengineering-12-00058-t006:** IL-10 levels in different states of RA (pg/mL).

States	Cases	Serum IL-10	SF IL-10	Synovium IL-10
Conditions of RA				
CON	3	7.09 ± 1.11	3.92 ± 0.72	55.80 ± 7.66
iCON	3	31.19 ± 3.52 ^a^	9.06 ± 1.95	162.01 ± 21.23
RA		57.12 ± 15.75 ^ab^	23.57 ± 7.64 ^ab^	414.00 ± 122.37 ^ab^
*F*-value		18.215	14.161	17.904
*p*-value		<0.001	<0.001	<0.001
Statuses of RA				
Disease remission	9	46.32 ± 10.16 ^ab^	27.87 ± 8.06 ^ab^	327.00 ± 89.44 ^ab^
Disease activity	9	67.92 ± 12.70 ^abc^	19.27 ± 4.19 ^abc^	501.00 ± 82.42 ^abc^
*F*-value		15.862	8.060	18.422
*p*-value		0.001	0.012	0.001
Stages of RA (disease remission status)				
early-stage	6	35.45 ± 4.63	20.27 ± 3.79	234.01 ± 26.50
moderate-stage	6	46.79 ± 5.23	26.60 ± 3.80	315.02 ± 23.93 ^A^
severe-stage	6	56.72 ± 4.96 ^AB^	36.73 ± 4.93 ^AB^	432.02 ± 31.53 ^AB^
*F*-value		13.896	11.684	39.319
*p*-value		0.006	0.009	<0.001
Stages of RA (disease activity status)				
early-stage	6	53.88 ± 5.67	17.27 ± 3.51	404.99 ± 32.61
moderate-stage	6	68.06 ± 4.99 ^A^	17.73 ± 3.67	522.04 ± 38.30 ^A^
severe-stage	6	81.81 ± 1.80 ^AB^	22.80 ± 4.03	576.05 ± 41.40 ^A^
*F*-value		29.108	2.019	16.197
*p*-value		0.001	0.214	0.004

^a^: *p* < 0.05 versus CON. ^b^: *p* < 0.05 versus iCON. ^c^: *p* < 0.05 versus disease remission. ^A^: *p* < 0.05 versus early-stage. ^B^: *p* < 0.05 versus moderate-stage.

## Data Availability

The raw data supporting the conclusion of this article will be made available from the first author.

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
