# Peer review of "The Differential Expressions and Associations of Intracellular and Extracellular GRP78/Bip with Disease Activity and Progression in Rheumatoid Arthritis"

_bioengineering, 2025, doi:10.3390/bioengineering12010058_

Round 1
Reviewer 1 Report (Previous Reviewer 3)
Comments and Suggestions for Authors
I think this is a useful paper that shows new biomarker candidates for RA patients. Although much research is still needed, if we can detect RA before the disease progresses, we can start early treatment.
Author Response
Responses to Reviewer 1
I think this is a useful paper that shows new biomarker candidates for RA patients. Although much research is still needed, if we can detect RA before the disease progresses, we can start early treatment.
Response. We would like to express our gratitude to the reviewers for acknowledging our contributions to this field of research.
RA is a complex systemic disease characterized by gradual advancement and deterioration as development in the stage of disease progression and the status of disease activity. Understanding the pathogenesis and progression of RA requires an analysis of both systemic and local immune responses, which are intricately linked. GRP78/BiP, a stress-induced protein and autoantigen in rheumatoid arthritis (RA), exhibits different expressions in various biological fluids and tissues, namely blood, synovial fluid (SF), and synovium, all of which are pertinent to the disease activity and progression of RA. However, the current research on GRP78/Bip is fragmented and lacks a comprehensive perspective, as existing studies on RA primarily focus on the role and targeted intervention of GRP78/Bip in either serum, SF, or synovium, without considering disease activity status and progression stage. This limited literature overlooks the systemic complexity of disease development and the changes that occur in RA.
This research indicated that GRP78/Bip present both intracellularly and extracellularly could serve as promising biomarkers for disease severity, as they effectively distinguished between individuals with and without RA, as well as different stages of disease progression among RA patients. Furthermore, intracellular and extracellular GRP78/Bip were able to differentiate between RA patients experiencing an activity or a remission of their symptoms. This suggests that GRP78/Bip could be valuable in the diagnosis and monitoring of RA. We believe that the future application of intracellular and extracellular GRP78/Bip in RA clinical settings could potentially help to predict which patients may have the disease and how severe the disease will be, as well as whether a patient in remission may experience an exacerbation of symptoms, and vice versa. All this would help improve RA management and overall quality of life.
Intracellular GRP78/Bip levels could be influenced by inflammatory mediators and have been linked to disease severity. On the other hand, extracellular GRP78/Bip levels may play a role in the immune response and tissue damage seen in RA. Understanding the differential expressions and associations of both intracellular and extracellular GRP78/Bip could provide insights into the pathogenesis of RA and potentially lead to new therapeutic targets for the disease. This understanding is crucial, as it can inform medical professionals about potential biomarkers for disease management and progression monitoring. Moreover, it highlights the need for further investigation into the mechanisms underlying these expressions, which could ultimately lead to improved therapeutic strategies tailored to individual patient needs.
This study provides a comprehensive analysis of the association between GRP78/Bip proteins and disease activity and progression of RA, hoping to pave the way for precision medicine strategies tailored to the specific disease condition of each patient, and overcome the limitations inherent in the conventional "one-size-fits-all" treatment models currently used for RA patients. While this study was the first to explore the differential expressions and significance of intracellular and extracellular GRP78/Bip in relation to disease activity and progression of RA, and points to the significance of subject-specific changes of GRP78/Bip in serum, SF, and synovium, additional molecular cell research, animal or in vitro experiments, and clinical studies are necessary to evaluate the effects of GRP78/Bip so as to provide further comprehensive evidence and guidance for its diagnostic value and clinical application in RA.

Reviewer 2 Report (New Reviewer)
Comments and Suggestions for Authors
The manuscript of Liu and colleagues reported an interesting analysis of GRP78/Bip and its role in the activity and progression of rheumatoid arthritis.
In my opinion the topic is very interesting and the research and the document are well done, but I think there are some things to review, others to explain better and I would like to see a new graph inserted asking you for an explanation.
- Line 32 explicit DAS 28 CRITERION
-Line 34 who are CON and iCON groups?
-Line 306 Please explain better the correlation between the figure and the table and between figure 1A and figure 1B where RA should represent the sum of the various groups in remission and RAaS of the groups in activity, it is not so intuitive.
-Line 332 blank
- Align the significances in the tables to make it more readable, I like how you wrote it in tables 2 and 3.
-In Table 2, the patients went from 16 and 48 in Table 1 to 3 and 9, respectively. What were the criteria for selecting patients with regard to the values measured for serum?
-WESTERN BLOTS. Who do the samples correspond to? Are they pools of samples or are they single samples from the 3 or 9 patients analyzed in Table 2? Please also explain this topic in the text and if they are from single patients, make sure that it is possible to understand which patients they are from, so as to compare the Elisa values with those of the lysate.
-CASPASE 3. what are you showing me is it a procaspase? a cleaved one? at what kDa are the bands you are showing me? Even in the originals you did not show me the kDa. If it is a procaspase an increase does not justify apoptosis if I do not see the cleaved form. Obviously even the cleaved form would not be enough, because other correlation analyses would be needed but at least it is an indication.
-line 405 blank
-Are the patients chosen for TNF-a and IL-10 analyses the same as those chosen for the analyses reported in Table 2?
-you showed me how the three factors analyzed increase during the progression of rheumatoid arthritis compared to other inflammatory diseases considered as controls. What happens, however, in a healthy patient? What are the basal levels? I understand that it is difficult to find some samples, but on the blood, what are the basal levels?
-I would like to see a comparison graph between GRP78/Bip, TNFα and IL-10 that shows how these factors are elevated in certain conditions. In the perspective of proposing GRP78/Bip as a marker of rheumatoid arthritis, it could be interesting to see that its values are significantly higher than TNFα and IL-10 which are the principles of inflammation. It would be interesting to delve deeper into this topic and investigate it by correlating it with healthy patients, and evaluate if this is really the case in the blood... understand if they are not present or are very low in the blood and then find it high in the pathology and in relation to the progression.
Thanks a lot for your work
Author Response
Responses to Reviewer 2
The manuscript of Liu and colleagues reported an interesting analysis of GRP78/Bip and its role in the activity and progression of rheumatoid arthritis.
In my opinion the topic is very interesting and the research and the document are well done, but I think there are some things to review, others to explain better and I would like to see a new graph inserted asking you for an explanation.
- Line 32 explicit DAS 28 CRITERION
Response. Thank you for point this out. The complete name of DAS 28 CRITERION has been clarified.
-Line 34 who are CON and iCON groups?
Response. The group designated as the control consisted of TMI Patients (CON), while the group for inflammation control included OA Patients (iCON). The sections that have been revised are highlighted in red within the paper.
-Line 306 Please explain better the correlation between the figure and the table and between figure 1A and figure 1B where RA should represent the sum of the various groups in remission and RAaS of the groups in activity, it is not so intuitive.
Response. Thank you for point this out.
Current research on rheumatoid arthritis (RA) remains fragmented and lacks a comprehensive perspective. Existing studies primarily focus on specific aspects of RA, such as the occurrence and diagnostic purposes of RA conditions, the statuses of RA in terms of disease activity or remission, and the stages of RA related to disease progression. However, the combined influence of these factors during the occurrence and development of RA is not fully considered. Simultaneously, the terminology issues surrounding the pathological processes of RA are significant. As we explored these concepts, it became apparent that considerable inconsistencies exist in the terminology employed within the field. In response to this challenge, we conducted this comprehensive study aimed at standardizing the terminology related to these factors. Our objective was to establish a clear and consistent framework that would serve as a valuable reference for scholars and researchers in future investigations. By addressing these terminological discrepancies, we hope to facilitate more effective communication and understanding within the academic community.
In our perspective, the onset and development of RA can be understood through three crucial phases: disease occurrence, disease activity and disease progression, which correspond to three states in RA: the conditions of RA, the statuses of RA, and the stages of RA. The conditions of RA are associated with the likelihood of disease occurrence. The statuses of RA relate to disease activity or remission that patients may experience throughout their illness, while the stages of RA are linked to the disease progression over time. Understanding RA in this way allows for a more comprehensive approach to treatment strategies and patient management.
The descriptions of the figures and tables have been enhanced in the revised manuscript.
-Line 332 blank
Response. Thank you for highlighting the absence of content on line 332. We have made the necessary modifications.
- Align the significances in the tables to make it more readable, I like how you wrote it in tables 2 and 3.
Response: Thank you for highlighting this. In response to your valuable feedback, we have made the necessary modifications to the descriptions of the tables in the revised manuscript. The alignment of the significance indicators within the tables has been adjusted to enhance overall readability.
-In Table 2, the patients went from 16 and 48 in Table 1 to 3 and 9, respectively. What were the criteria for selecting patients with regard to the values measured for serum?
Response. Thank you for point this out.
Table 2 ultimately comprises 128 patients (68.82%), with 16 patients allocated to each of the following groups: CON, iCON, aES RA, aMS RA, aSS RA, rES RA, rMS RA, and rSS RA groups. All the enrolled 128 patients were tested for serum GRP78/Bip. Initially, serum GRP78/Bip levels were assessed in all 128 enrolled patients, as serum test results are more readily obtainable. Following this, patients were then selected for further studies based on the monitoring outcomes of serum GRP78/Bip.
Despite continued enrollment over a three-year period, only 50 patients proceeded to surgical intervention. Subsequently, participants from each group were further enrolled in a 1:1 ratio, matched according to gender. Within the same group, patients were subsequently screened based on their median serum levels of GRP78/Bip. Ultimately, 24 patients were involved, with 3 patients from each group included in the follow-up study.
The RA remission status (9 patients) involved early-stage (rES RA, 3 patients), moderate-stage (rMS RA, 3 patients) and severe-stage (rSS RA, 3 patients) patients, similarly, the RA activity status (9 patients) including early-stage (aES RA, 3 patients), moderate-stage (aES RA, 3 patients) and severe-stage (rES RA, 3 patients).
-WESTERN BLOTS. Who do the samples correspond to? Are they pools of samples or are they single samples from the 3 or 9 patients analyzed in Table 2? Please also explain this topic in the text and if they are from single patients, make sure that it is possible to understand which patients they are from, so as to compare the Elisa values with those of the lysate.
Response. Thank you for bringing this to my attention.
The limited sample size of pathological synovial tissue presents a challenge, as the total protein content derived from individual samples is insufficient for comprehensive western blot analysis. To address this issue, this study employed a strategy whereby tissue samples from three different patients within the same group were combined to create a sample pool. This approach resulted in the formation of distinct sample cohorts, specifically the CON group, iCON group, aES RA group, aMS RA group, aSS RA group, rES RA group, rMS RA group, and rSS RA group. We seek to enhance the overall representativeness and reliability of the data by merging samples into sample pools. This approach allows for a more robust analysis that mitigates potential biases that may arise from isolated samples. In particular, it addresses the issue of samples that may go undetected due to low protein content, which can lead to gaps in the data. Moreover, by pooling samples, the researchers facilitate a more effective comparison between their findings and the results obtained from ELISA techniques, thereby strengthening the validity of their conclusions. Additionally, to ensure the accuracy and validity of the experimental measurements, the biological samples from each group underwent assessment three times in parallel. This methodological rigor involved performing multiple technical repetitions for each sample, from which an average value was subsequently calculated. The resulting analyses allowed for the generation of graphical representations that effectively conveyed the outcome of the experiments, thus providing a clearer illustration of the findings derived from the pooled samples.
This topic is discussed in the manuscript, and the revisions made are indicated in red within the paper.
-CASPASE 3. what are you showing me is it a procaspase? a cleaved one? at what kDa are the bands you are showing me? Even in the originals you did not show me the kDa. If it is a procaspase an increase does not justify apoptosis if I do not see the cleaved form. Obviously even the cleaved form would not be enough, because other correlation analyses would be needed but at least it is an indication.
Response. In response to these concerns, we would like to express our gratitude for highlighting this issue.
Caspase-3 is widely recognized as a fundamental indicator of apoptosis and is regarded as an essential element in the apoptotic signaling pathway. This enzyme plays a role, either fully or partially, in the cleavage of various essential proteins that are involved in the process of programmed cell death. Its activity can either partially or entirely facilitate the breakdown of these essential proteins, thereby influencing the progression and execution of apoptosis. This function underscores the importance of caspase-3 content in the regulation of cell fate and its potential implications in various physiological and pathological contexts.
Full-length Caspase-3 typically exists as a 32 kDa proenzyme. This form is essential for the regulation of apoptosis, a process of programmed cell death. During apoptosis, Caspase-3 undergoes a precise cleavage that results in the formation of two distinct fragments: a 17 kDa fragment (p17) and a 12 kDa fragment (p12), which are involved in executing the final stages of the apoptotic process.
Cleaved caspase-3 serves as a key biomarker that is exclusively found in cells undergoing apoptosis, thus providing a clear distinction from non-apoptotic or necrotic cells, which do not exhibit the presence of this protein. However, the levels of cleaved caspase-3 are low and challenging to detect within pathological synovial tissue. Consequently, the utilization of cleaved caspase-3-specific antibodies is particularly advantageous in experimental setups that aim to identify apoptotic cells within a mixed population. Techniques such as immunohistochemistry (IHC), immunofluorescence (IF), and flow cytometry are well-suited for this purpose, as they facilitate the observation and analysis of apoptotic cells more effectively.
In applications related to western blotting (WB), our initial belief was that the full-length caspase-3 antibody would serve as the optimal selection for detection purposes. By assessing the levels of full-length caspase-3, we can glean valuable information regarding the dynamics of apoptosis; indeed, any fluctuations in the content of full-length caspase-3 can provide significant indications of the progression or modulation of this critical biological process. In our prior research efforts, our team utilized full-length Caspase-3 to identify tissue proteins through Western blot (WB) techniques. Furthermore, for a variety of other analytical methods—including immunohistochemistry (IHC), immunofluorescence (IF), and flow cytometry analysis—we routinely relied on cleaved caspase-3. However, the reviewer's insights are more accurate and serve as important guidance for our future research. We are grateful for bringing this matter to our attention. This approach allowed us to gain valuable insights into the functional state of the proteins and their roles in various biological processes.
In the present study, however, the limited quantity of pathological synovial tissue samples hindered the performance of any immunohistochemistry or immunofluorescence experiments. The scarcity of these tissue samples restricts the range of analytical techniques that can be employed, which in turn limits the depth and breadth of the research findings. As a result, the inability to conduct these crucial experiments underscores a critical constraint in exploring the complexities of the pathological conditions being studied.
Given the varied formatting requirements of each journal, we labeled only the name of the target protein in our submission, omitting the protein kDa. In the revised manuscript, the kDa for Caspase-3 has now been included.
-line 405 blank
Response. Thank you for highlighting the absence of content on line 405. We have made the necessary modifications.
-Are the patients chosen for TNF-a and IL-10 analyses the same as those chosen for the analyses reported in Table 2?
Response. The patients selected for the analyses of TNF-α and IL-10 (refer to Tables 5 and 6) are identical to those used in the analyses presented in Table 3, rather than in Table 2.
-you showed me how the three factors analyzed increase during the progression of rheumatoid arthritis compared to other inflammatory diseases considered as controls. What happens, however, in a healthy patient? What are the basal levels? I understand that it is difficult to find some samples, but on the blood, what are the basal levels?
Response. Thank you for point this out.
One of the limitations identified in this paper is the necessity for further research to determine the basal levels of GRP78/Bip. “Furthermore, the normal levels of GRP78/Bip in serum, SF, and synovium have not been definitively established, nor have we fully understood the significance of these levels. Various factors including age, sex, diet, ethnicity, physical activity, medication use, the method of estimating GRP78/Bip, and sample storage conditions all have the potential to influence GRP78/Bip levels. Further research is needed to establish a baseline for normal GRP78/Bip levels, understand its implications, and account for potential confounding variables that could affect measurements.”
Currently, there is a lack of consensus concerning the baseline levels of GRP78/Bip, TNF-α, and IL-10 in healthy individuals. This discrepancy can be attributed to various factors, including the diversity of detection methods employed, variations in detection techniques, as well as the inherent heterogeneity among individuals. These differences can lead to significant variability in the reported levels of these biomarkers. Addressing this variability is an important focus for future research, aiming to establish standardized measurements and better understand the underlying mechanisms that contribute to these variations.
-I would like to see a comparison graph between GRP78/Bip, TNFα and IL-10 that shows how these factors are elevated in certain conditions. In the perspective of proposing GRP78/Bip as a marker of rheumatoid arthritis, it could be interesting to see that its values are significantly higher than TNFα and IL-10 which are the principles of inflammation. It would be interesting to delve deeper into this topic and investigate it by correlating it with healthy patients, and evaluate if this is really the case in the blood... understand if they are not present or are very low in the blood and then find it high in the pathology and in relation to the progression.
Response. Thank you for point this out.
The limited enrollment of patients in the study hindered the ability to conduct a comparison graph between GRP78/Bip, TNFα and IL-10 that shows how these factors are elevated under certain conditions. Future research should incorporate a larger sample size to introduce more significant findings. The relationship between GRP78/Bip, TNFα, and IL-10 in the context of rheumatoid arthritis (RA) has not yet been thoroughly confirmed. It is essential to conduct more comprehensive prospective studies to establish the nature of these potential causalities. Longitudinal or cohort studies could provide valuable insights and help to clarify how these factors interact within the disease framework. Such research is vital for advancing our understanding of RA and developing effective interventions.
Due to the small number of patients with rheumatoid arthritis undergoing surgical treatment, the amount of Synovium sample are difficult to collect. In our future study, we plan to study the correlation between GRP78/Bip (serum or SF) and the disease occurrence, disease activity and disease Progression of RA, and analyze the Logistic regression study between GRP78/Bip (serum or SF) and the disease occurrence, disease activity and disease Progression of RA. ROC analysis was performed on whether RA occurred, whether RA was active or not, and the disease progression of RA. Subgroup analysis of RA occurrence, RA disease activity, and RA disease progression was performed according to the optimal cut-off value of GRP78/Bip in serum or joint fluid obtained by ROC curve analysis
The limited number of patients diagnosed with rheumatoid arthritis (RA) who undergo surgical interventions presents a significant challenge in collecting a sufficient amount of Synovium samples. This scarcity hampers extensive research efforts. In our forthcoming study, we aim to investigate the relationships between the levels of GRP78/Bip, found in either serum or synovial fluid, and various aspects of rheumatoid arthritis (RA), including disease occurrence, disease activity, and disease progression of RA. We will conduct a logistic regression analysis to discern how GRP78/Bip levels correlate with these critical facets of the disease (RA occurrence, RA activity, and RA progression). Moreover, we will utilize receiver operating characteristic (ROC) analysis to assess the likelihood of RA occurrence, identify whether the RA is in an active state, and evaluate the disease progression of RA. Furthermore, a subgroup analysis regarding the disease occurrence, disease activity, and disease progression of RA was conducted based on the optimal cut-off values of GRP78/Bip in serum or SF as determined by ROC curve analysis. This comprehensive approach will enhance our understanding of the role GRP78/Bip plays in the pathology of RA.
The identification of GRP78/BiP-specific autoantibodies has emerged as a novel diagnostic tool, potentially serving as both a preclinical indicator and an improved diagnostic biomarker for RA. Recent integrative studies evaluating the diagnostic value of GRP78/BiP and anti-GRP78/BiP antibodies in saliva, blood, SF, and synovium for RA has demonstrated a high pooled specificity of 0.92 and a moderate pooled sensitivity of 0.67. The overall diagnostic odds ratio (DOR) was notably high at 23.73, while the pooled positive likelihood ratio (LR+) was 8, indicating that individuals diagnosed with RA are eight times more likely to test positive for biomarkers associated with GRP78/BiP compared to those without the condition. Consequently, GRP78/BiP-specific autoantibodies could serve as a valuable supplement to current diagnostic strategies, thereby enhancing diagnostic precision and contribute to more informed clinical decision-making.
The data from the 'Duisseldorf Rheumaregister' retrieval study included analysis of 277 RA patients and 893 patients with other rheumatic diseases to identify the presence of RF. The anti-GRP78/BiP antibody demonstrated equal sensitivity (66% vs 68%) and higher specificity (99% vs 76%) compared to the RF. It is important to note that 5% of the apparently healthy population also yield positive results for RF. The high specificity of GRP78/BiP or anti-GRP78/BiP antibodies significantly exceeded that of RF based on research findings. Unlike RF, which targets the Fc portion of IgG and formulates the immune complexes that exceed the process of RA as well as other rheumatoid diseases, GRP78/BiP antibodies exhibited a more focused immune response. This enhanced specificity was crucial in distinguishing between RA and other rheumatoid diseases. ACPAs have been identified as highly specific biomarkers for diagnosing RA and were believed to be closely linked to the pathogenesis of arthritis. Notably, the presence of anti-CCP antibodies could be detected several years before the onset of joint inflammation Given the high specificity of ACPAs in RA diagnosis, their role in the pathogenesis of RA has become the focus of active investigation. Similarly, GRP78/BiP or anti-GRP78/BiP antibodies showed comparable sensitivity to ACPAs (67% vs. 67%) and slightly lower specificity (92% vs. 95%). This resemblance may stem from their similar role in the pathogenesis of RA. With moderate sensitivity and high specificity, GRP78/BiP or anti-GRP78/BiP antibodies could serve as a valuable supplement to existing diagnostic methods. Given their distinct roles in RA pathogenesis, a combination of GRP78/BiP and anti-GRP78/BiP antibodies with RF or ACPAs could provide even more reliable and accurate results for the clinical diagnosis of RA. Their different roles in the pathogenesis of RA made them complementary to each other, enhancing the overall effectiveness of diagnostic testing. The combination of anti-GRP78/BiP antibodies and anti-CCP antibodies could obtain higher specificity than that of anti-CCP antibodies detected individually. This tandem combination approach has demonstrated good clinical value in differentiating RA from other autoimmune diseases. These findings highlighted the importance of considering multiple biomarkers and combined detection in diagnostic testing for RA, as the presence of both anti-GRP78/BiP and anti-CCP antibodies together could provide more accurate and specific results.
Research on potential clinical biomarkers in RA has been a prominent topic over the past two decades, yielding numerous significant findings. The investigation of GRP78/BiP-specific and anti-GRP78/BiP-specific antibodies has been conducted by comparing levels in healthy individuals, assessing their presence in the bloodstream. In contrast, these antibodies are found to be absent or present at significantly low levels in the blood, while their concentration is elevated in rheumatoid arthritis (RA) pathology. These results have been documented in scientific literature. Nevertheless, the connection between GRP78/BiP-specific or anti-GRP78/BiP-specific antibodies and the disease progression of RA has not yet been explored thoroughly. This represents an area for future research.

Reviewer 3 Report (New Reviewer)
Comments and Suggestions for Authors
This manuscript focuses on the use of GRP78 as a biomarker readout to assess RA stage in blood and synovium/synovial fluid. The article appears to be well thought out and does have merits to the field. Effort was made to support the premise of the study and the discussion using adequate citations. Unfortunately, there are some technical aspects to the work that would need addressed prior to this reviewer reccomending it for publication. This reviewer believes that if these issues are corrected then a revised manuscript would be suitable for publication. The following issues are noted:
1. The use of simple caspase 3 expression is not an accepted method to evaluate apoptosis. Caspase 3 blots in the context of apoptosis should include both cleaved and full length caspase, as cleaved caspase will both signify activation of the caspase as well as inform the reader if changes in full length caspase are a result of lower expression or increased cleavage. The images in the paper and full length blots are cut so close to the unspecified immunopositive band that one cannot determine if it is full length or cleaved and what the status is on the protein in the context of apoptosis.
2. The methods section is not adequately specific enough to allow other researchers to replicate the work. Issues include:
a. Vagueness- were proteins truly run on 30% acrylamide gels, or was the stock acrylamide 30%?
b. Improper procedure- Were bands actually excised from the nitrocellulose membrane prior to imaging? And what purpose does that serve?
c. Vagueness- CST offers 5 caspase antibodies, each detecting different combinatiions of cleaved and full length protein. Please include catlaog numbers and the specific targets.
d. Possible miscommunication of method- CST does not appear to manufacture or offer an imager to detect chemiluminescence. Please prvide a catalog number, or alternatively provide an accurate description of the work performed and the equipment used to perform the work.
e. Unacceptable explanation of statistical testing- What post hoc tests were performed to complete the multiple comparisons? What software was used?
3. Figure legends do not contain required information such as: what does data represent? What is the n? What statistical tests were used for the panels? Figures should contain enough information to allow them to be interpreted without hunting through the text to find information.
4. Figure 5: What software was used to generate this figure? What do the solid and dashed lines represent?
Author Response
Responses to Reviewer 3
This manuscript focuses on the use of GRP78 as a biomarker readout to assess RA stage in blood and synovium/synovial fluid. The article appears to be well thought out and does have merits to the field. Effort was made to support the premise of the study and the discussion using adequate citations. Unfortunately, there are some technical aspects to the work that would need addressed prior to this reviewer reccomending it for publication. This reviewer believes that if these issues are corrected then a revised manuscript would be suitable for publication. The following issues are noted:
- The use of simple caspase 3 expression is not an accepted method to evaluate apoptosis. Caspase 3 blots in the context of apoptosis should include both cleaved and full length caspase, as cleaved caspase will both signify activation of the caspase as well as inform the reader if changes in full length caspase are a result of lower expression or increased cleavage. The images in the paper and full length blots are cut so close to the unspecified immunopositive band that one cannot determine if it is full length or cleaved and what the status is on the protein in the context of apoptosis.
Response. In response to these concerns, we would like to express our gratitude for highlighting this issue.
Caspase-3 is widely recognized as a fundamental indicator of apoptosis and is regarded as an essential element in the apoptotic signaling pathway. This enzyme plays a role, either fully or partially, in the cleavage of various essential proteins that are involved in the process of programmed cell death. Its activity can either partially or entirely facilitate the breakdown of these essential proteins, thereby influencing the progression and execution of apoptosis. This function underscores the importance of caspase-3 content in the regulation of cell fate and its potential implications in various physiological and pathological contexts.
Full-length Caspase-3 typically exists as a 32 kDa proenzyme. This form is essential for the regulation of apoptosis, a process of programmed cell death. During apoptosis, Caspase-3 undergoes a precise cleavage that results in the formation of two distinct fragments: a 17 kDa fragment (p17) and a 12 kDa fragment (p12), which are involved in executing the final stages of the apoptotic process.
Cleaved caspase-3 serves as a key biomarker that is exclusively found in cells undergoing apoptosis, thus providing a clear distinction from non-apoptotic or necrotic cells, which do not exhibit the presence of this protein. However, the levels of cleaved caspase-3 are low and challenging to detect within pathological synovial tissue. Consequently, the utilization of cleaved caspase-3-specific antibodies is particularly advantageous in experimental setups that aim to identify apoptotic cells within a mixed population. Techniques such as immunohistochemistry (IHC), immunofluorescence (IF), and flow cytometry are well-suited for this purpose, as they facilitate the observation and analysis of apoptotic cells more effectively.
In applications related to western blotting (WB), our initial belief was that the full-length caspase-3 antibody would serve as the optimal selection for detection purposes. By assessing the levels of full-length caspase-3, we can glean valuable information regarding the dynamics of apoptosis; indeed, any fluctuations in the content of full-length caspase-3 can provide significant indications of the progression or modulation of this critical biological process. In our prior research efforts, our team utilized full-length Caspase-3 to identify tissue proteins through Western blot (WB) techniques. Furthermore, for a variety of other analytical methods—including immunohistochemistry (IHC), immunofluorescence (IF), and flow cytometry analysis—we routinely relied on cleaved caspase-3. However, the reviewer's insights are more accurate and serve as important guidance for our future research. We are grateful for bringing this matter to our attention. This approach allowed us to gain valuable insights into the functional state of the proteins and their roles in various biological processes.
In the present study, however, the limited quantity of pathological synovial tissue samples hindered the performance of any immunohistochemistry or immunofluorescence experiments. The scarcity of these tissue samples restricts the range of analytical techniques that can be employed, which in turn limits the depth and breadth of the research findings. As a result, the inability to conduct these crucial experiments underscores a critical constraint in exploring the complexities of the pathological conditions being studied.
- The methods section is not adequately specific enough to allow other researchers to replicate the work. Issues include:
- Vagueness- were proteins truly run on 30% acrylamide gels, or was the stock acrylamide 30%?
Response. Thank you for point this out.
This manuscript contains excessive wording, and since the Western Blot (WB) experiment is a standard procedure, it has been abbreviated in this document.
Equal amounts of total protein from synovium lysates or SF samples were separated by 10% sodium dodecyl sulphate-polyacrylamide gel electrophoresis (SDS-PAGE) (1.5M TRIS·Hcl, pH = 8.8, 30% acrylamide, 10 % SDS, AP, TEMED), and then transferred onto nitrocellulose membranes (Millipore, Billerica, MA, USA).
|
Concentration |
|||||||||||
Reagent |
8% |
10% |
12% |
15% |
18% |
20% |
8% |
10% |
12% |
15% |
18% |
20% |
H2O (ml) |
4.63 |
4 |
3.3 |
2.3 |
1.3 |
0.63 |
6.9 |
5.9 |
4.9 |
3.4 |
1.9 |
0.9 |
30% acrylamide(29:1) (ml) |
2.67 |
3.3 |
4 |
5 |
6 |
6.67 |
4 |
5 |
6 |
7.5 |
9 |
10 |
1.5M TRIS.Hcl(PH 8.8) (ml) |
2.5 |
2.5 |
2.5 |
2.5 |
2.5 |
2.5 |
3.8 |
3.8 |
3.8 |
3.8 |
3.8 |
3.8 |
10%SDS (ml) |
0.1 |
0.1 |
0.1 |
0.1 |
0.1 |
0.1 |
0.15 |
0.15 |
0.15 |
0.15 |
0.15 |
0.15 |
AP (ml) |
0.1 |
0.1 |
0.1 |
0.1 |
0.1 |
0.1 |
0.15 |
0.15 |
0.15 |
0.15 |
0.15 |
0.15 |
TEMED (ul) |
5ul |
5ul |
5ul |
5ul |
5ul |
5ul |
7.5ul |
7.5ul |
7.5ul |
7.5ul |
7.5ul |
7.5ul |
Total volume (ml) |
10ml |
15ml |
- Improper procedure- Were bands actually excised from the nitrocellulose membrane prior to imaging? And what purpose does that serve?
Response. Thank you for point this out.
The whole membrane was cut to blot for different antibodies. Following the Western Blot (WB) experiment, the protein bands from the whole membrane were carefully excised to preserve only the target protein. In the case of analyzing a whole film, it is essential to perform a complete stripping process. This involves incubating with a second or potentially a third antibody, which can be quite time-consuming. When the conditions for stripping are excessively harsh or when multiple rounds of elution are conducted, there is a risk that the antigens previously transferred to the entire membrane surface may also be removed. This can lead to a reduction in the final signal observed in Western Blot analysis. Furthermore, if the stripping buffer is not adequately washed away, remaining components can interfere with the binding efficiency of the subsequent antibodies, complicating the results.
- Vagueness- CST offers 5 caspase antibodies, each detecting different combinatiions of cleaved and full length protein. Please include catlaog numbers and the specific targets.
Response. Thank you for point this out.
There was some confusion regarding the experimental methods employed, which inadvertently overlapped with those utilized in our previous research. This oversight resulted from insufficient review on our part. However, in light of the reviewer's constructive feedback, we have thoroughly re-examined the experimental methodology presented throughout the entire paper to ensure clarity and accuracy.
Specific targets |
Brand |
Catalog number |
Caspase 3 |
Abways |
CY5384 |
CRP78/Bip |
Proteintech |
11587-1-AP |
GAPDH |
Servicebio |
GB11002 |
Goat anti-Rabbit IgG(H+L) |
Servicebio |
G1213 |
- Possible miscommunication of method- CST does not appear to manufacture or offer an imager to detect chemiluminescence. Please prvide a catalog number, or alternatively provide an accurate description of the work performed and the equipment used to perform the work.
Response. Thank you for point this out.
The band was excised for protein identification using a chemiluminescence detection system from Shenhua Science Technology Co., Ltd, instead of CST Corporation, as originally stated in the manuscript. This discrepancy was a writing error in the original manuscript. There was some confusion regarding the experimental methods employed, which inadvertently overlapped with those utilized in our previous research. This oversight resulted from insufficient review on our part. However, in light of the reviewer's constructive feedback, we have thoroughly re-examined the experimental methodology presented throughout the entire paper to ensure clarity and accuracy.
The catalog number of the chemiluminescence detection system (Shenhua Science Technology Co., Ltd., Hangzhou, China) is HZAN5Q, and is identified by the model number SH-Compact523.
- Unacceptable explanation of statistical testing- What post hoc tests were performed to complete the multiple comparisons? What software was used?
Response. Thank you for point this out.
- Figure legends do not contain required information such as: what does data represent? What is the n? What statistical tests were used for the panels? Figures should contain enough information to allow them to be interpreted without hunting through the text to find information.
Response. Thank you for point this out.
The value of the data is represented by the vertical axis of the graphs, and the revised manuscript includes the statistical tests and the variable "n."
- Figure 5: What software was used to generate this figure? What do the solid and dashed lines represent?
Response. Thank you for point this out.
The figure presented in this study was generated using Origin 8.5 software (OriginLab Corporation, Northampton, Massachusetts, USA). Specifically, the "Line + Symbol" feature within Origin was utilized to create a point plot. This type of plot effectively illustrates both the individual data points and the lines (solid or dashed lines) that connect them, providing a clear visual representation of the data. Such charts are frequently employed to showcase experimental findings, as they allow for a simultaneous examination of the distribution of data points and the overall trend observed within the dataset.

Round 2
Reviewer 2 Report (New Reviewer)
Comments and Suggestions for Authors
Thanks to Liu and colleagues for their responses to my revisions of their manuscript. In this form the manuscript sounds good and I accept it in its current form.
Author Response
Thanks to Liu and colleagues for their responses to my revisions of their manuscript. In this form the manuscript sounds good and I accept it in its current form.
Response. we wish to convey our heartfelt appreciation to the reviewers for the recognition of our contributions to this area of research. Their acknowledgment serves as a valuable affirmation of our efforts, and we are grateful for the constructive feedback that has helped refine our work.

Reviewer 3 Report (New Reviewer)
Comments and Suggestions for Authors
Thank you for your responses. The updated manuscript still suffers from a limitation on the caspase 3 / cleaved caspase 3 findings, however this reviewer understands the limited availibility of samples and the author's inability to correct this limitation. This reviewer would be more enthusiastic if this limitation were acknowledged, which could be simply be addressed by a statement that "additional work (or future work) examining apoptosis and caspase cleavage is warrented based on our results" or something to that effect.
Despite this shortcoming the article is now improved, and I will suggest that it be accepted with a minor concession on the caspase acknowledgement. Thank you.
Author Response
Thank you for your responses. The updated manuscript still suffers from a limitation on the caspase 3 / cleaved caspase 3 findings, however this reviewer understands the limited availibility of samples and the author's inability to correct this limitation. This reviewer would be more enthusiastic if this limitation were acknowledged, which could be simply be addressed by a statement that "additional work (or future work) examining apoptosis and caspase cleavage is warrented based on our results" or something to that effect.
Despite this shortcoming the article is now improved, and I will suggest that it be accepted with a minor concession on the caspase acknowledgement. Thank you.
Response. Thank you for point this out. This limitation has been recognized and can be addressed in the revised manuscript with a statement indicating that "The limited availability of pathological synovium samples constrains the variety of analytical methods that can be employed, including immunohistochemistry and immunofluorescence. This scarcity further restricts the comprehensiveness of research findings, potentially impacting the overall conclusions drawn from the data. Further investigation into apoptosis and cleaved caspases is warranted, as indicated by our findings."

This manuscript is a resubmission of an earlier submission. The following is a list of the peer review reports and author responses from that submission.
Round 1
Reviewer 1 Report
Comments and Suggestions for Authors
Comment
The manuscript deals with differential expression and associations of intracellular and extracellular GRP78/Bip with RA disease activity and RA progression. Unfortunately, the manuscript, even though the manuscript contains very detailed descriptions, the specific topic and results are very incomprehensible presented. Already in the introduction, many facts concerning RA from the literature are stated, but these facts do partly not show the connection with the specific topic. The introduction is overloaded with facts. Unfortunately, the manuscript in its present form does not meet the standard for publication in Bioengineering.
Reviewer 2 Report
Comments and Suggestions for Authors
The manuscript entitled ‘The Differential Expressions and Associations of Intracellular and Extracellular GRP78/Bip with Disease Activity and Progression in Rheumatoid Arthritis’ by Guoyin Liu et al., gives evidence that expression levels of GRP78/Bip, TNF-α, and IL-10 were elevated in serum, SF, and synovium of patients with RA when compared to both the CON and iCON groups. In serum and synovium of patients with RA, positive correlations were observed between the levels of GRP78/Bip expression and the stages of disease progression, along with TNF-α and IL-10 levels. Additionally, in SF, positive correlations were identified between the expression levels of GRP78/Bip and the concentrations of TNF-α and IL-10; conversely, a negative correlation was noted between GRP78/Bip expression and stages of disease progression.The authors conclude that differential expressions of GRP78/Bip in serum, SF, and synovium indicated that the potential role and function of GRP78/Bip might vary depending on its location within the specific biological fluids and tissues. The presence of intracellular and extracellular GRP78/Bip was associated with disease activity and progression of RA, suggesting the involvement of GRP78/Bip in the pathogenesis and development of this debilitating autoimmune disorder, as well as its potential as a biomarker for monitoring disease activity and progression.
In general, the manuscript is very confusing and needs work to straighten this issue, but the principle idea is very good.
Specific comments:
I really like the work submitted but would strongly recommend improvement. This is work to be done by senior scientists.
11. The Abstract is not clear and does not convince the reader of any good. Needs clarification.
22. The Introduction, especially the entire description of RA is lengthy and unprecise, please shorten substantially. Molecular mechanism of RA, in particular the ER lumen, is outlined adequately to help the reader find the point.
33. What do the authors mean with ‘GRP78/BiP can be overexpressed in salvia, synovium, serum and synovial fluid?’. Synovium is connective tissue, while salvia, serum and SF is liquid. Just on the side there is no serum in the human body only plasma. Serum is man made by coagulation. It’s simply not precise.
44. GRP78/BiP is a stress-induced protein and autoantigen, with intra-cellular and extracellular forms. The authors have selected this protein of it’s potential relevance in RA, but please let the reader know what you really mean and what you were looking at. For me it makes a difference whether you look at tissue ore fluid since a protein released in fluid is not the same as bound in the E>CM-network of connective tissue. We have this weakness throughout the entire paper.
55. Thus, the aim of this study was to investigate the differential expression of GRP78/Bip in serum, SF, and synovium, as well as to explore the association of intracellular and extracellular GRP78/Bip with disease activity and progression in patients diagnosed with RA. A better understanding of the function of intra-cellular and extracellular forms of GRP78/Bip, along with the mechanisms regulating GRP78/Bip-induced immune responses, may result in innovative strategies to the management of RA and possibly other inflammatory diseases.
66. Another issue, a study like this needs an approval from the ethic commission of the Nanjing Medical University, which is missing, and it is not enough to follow the Helsinki Declaration, since this only regulates human rights. Who is in charge of protecting the rights of the patients – authority?
7. Line 271: ‘Exclusion criteria for this study included a diagnosis of autoimmune diseases other than RA’. Diagnostic parameters like anti-nucleated antibodies for RA but which one for the other autoimmune diseases…. should be provided in the supplements.
87. 16 patients in each of the following groups: CON group, iCON group, aES RA 286 group, aMS RA group, aSS RA group, rES RA group, rMS RA group, and rSS RA group. How can the authors select exactly 16 patients for each group without excluding by purpose and induce bias?
98. Line 340: ‘SF samples were aspirated from the knee joint of patients prior to surgical procedures and extracted during capsular cutting in the surgical procedures. The samples were collected in heparinized containers to prevent coagulation and subsequent cell damage. What kind of heparin and in which concentration since heparin can interfere with immune cell function.
19. Line 344: What do the authors mean with ‘Diseased synovium samples’ since diseased means dead.
110. Lane 356: Western blotting: No information is provided on the polyacrylamide gels, are the precasted or serf-made no company is given, same is apparent for the nitrocellulose membrane, the reagent for no specific blocking (milk)
111. Results are too confusing for me, with too many groups and comparisons. Please work on clarity. Figure 1 also does not help to solve this problem, too much information no clarity. Figure legend could help.
112. Lane 404: ‘In essence, the aforementioned findings revealed that GRP78/Bip levels in serum of RA patients increased progressively as disease progresses’ (two times progress). The reviewer finds no use of these statements, since it would mean that groups are less important than a general trend in the disease course.
113. Lane 415: ‘Comprehending the role of GRP78/Bip in the pathogenesis and development of RA necessitates scrutinizing both systemic and local immune reactions that are closely connected’. The result should clearly outline the relevant findings without discussion. That can be done in the discussion section.
114. Figure Legend 3: Its hard to follow what is A+B as well as C+D. There is no indication.
115. Table 3.1/2/3 is overloaded with numbers and relevance is camouflaged.
116. Same is true for Table 4.1/2 and Table 5.1/2.
117. Discussion is a repetition line 545 to 592 of the introduction. Then a Biomarker relevance is discussed with no prelude.
118. Relevance starts with: ‘Multiple studies have provided evidence of significant differences in the functions of GRP78/BiP depending on its extracellular or intracellular location. This versatility allows GRP78/BiP to be classified as a true moonlighting protein, with two distinct roles in vivo[38]. The intracellular form of GRP78/BiP serves as a critical chaper one in protein folding within ER and possesses anti-apoptotic properties, aiding in cell protection under stressful conditions. Conversely, the extracellular form of GRP78/BiP exhibits potent immunomodulatory and anti-inflammatory effects, although the specific cell-surface receptor through which it acts remains unidentified, with expression being mainly on monocytes’. But this is not a discussion of relevance of the elaborated data sets of the authors, rather important issues that need to go to the introduction to prepare the readers for the upcoming topic.
219. Authors discuss technologies, some highly sophisticated, line 633 – 638 to recommend ELISA and Western blotting. Sometimes, keeping issues simple helps, but you cannot compare ELISA with mass spectrometry – sorry. Please discuss your data sets and focus on the important issues.
220. From the mass of irrelevant information provided, it is hard to separate results provided by the authors and those that are simply cited such as the following sentence. ‘Unlike healthy individuals and patients with other rheumatic diseases where measurable GRP78/BiP reactive T cells were undetectable, RA patients exhibited pronounced T cell reactivity to GRP78/BiP’. Which other rheumatic diseases and why should a T cell response be so specifically linked to GRP78/BiP in RA patients. Please complete discussion.
221. Next issue is as vague: ‘Since overexpression of GRP78/BiP has been demonstrated to reduce the sensitivity of cells to cytotoxic T cell-mediated killing, GRP78/BiP overexpression and GRP78/BiP specific autoimmunity may be involved in the pathogenesis of RA. Even more complicated, the authors switch to cytotoxic T cell reaction, again without further discussion. Here MHC Class I is involved, while in §21 we have a MHC Class II-dependent recognition when addressed from the immunological perspective.
222. Consequently, the detection of GRP78/BiP-specific autoantibodies has emerged as a novel diagnostic tool, potentially serving as both a preclinical indicator and an improved diagnostic biomarker for RA. Has nothing to do with a discussion of data generated in the study, rather we go back to ‘biomarker businesses. Please focus on the relevance of your finding.
223. Along the line the next paragraph now changes to ‘review style’. Now the topic is the 'Duisseldorf Rheumaregister' retrieval study and in this context the anticitrullinated protein/peptide antibodies (ACPAs) are discussed as biomarker and anti-CCP antibodies show up from nowhere. All that information is not different to the Introduction section and does not help a discussion to be started.
224. In general, the Discussion is a review – I stop commenting now– Discussion needs to be shortened by 70 % of the volume to focus on a discussion.
225. Conclusion is a sum-up of biomarker relevance and not at all visionaries with no critical limitations discussed. It’s like the repetition of the repetition of the repetition.
The language should be revised, and the manuscript shortened to 1/3 of the words to improve clear statements and suppress review style completions.
Comments on the Quality of English LanguageLanguage should be revised
Reviewer 3 Report
Comments and Suggestions for Authors
This is an interesting article in which GRP78/Bip is associated with RA disease activity and progression, suggesting its potential as a biomarker.
The diagram is difficult to understand, so please make some simple corrections.
Please add a brief description of your sample in each figure regends.
Please make the font of the sample name in Figure 2 - 4 larger.